# INFLUENCE ESTIMATION FOR GENERATIVE ADVERSARIAL NETWORKS

**Naoyuki Terashita**    **Hiroki Ohashi**    **Yuichi Nonaka**    **Takashi Kanemaru**

Hitachi, Ltd.
Tokyo, Japan

## ABSTRACT

Identifying harmful instances, whose absence in a training dataset improves model performance, is important for building better machine learning models. Although previous studies have succeeded in estimating harmful instances under supervised settings, they cannot be trivially extended to generative adversarial networks (GANs). This is because previous approaches require that (i) the absence of a training instance directly affects the loss value and that (ii) the change in the loss directly measures the harmfulness of the instance for the performance of a model. In GAN training, however, neither of the requirements is satisfied. This is because, (i) the generator's loss is not directly affected by the training instances as they are not part of the generator's training steps, and (ii) the values of GAN's losses normally do not capture the generative performance of a model. To this end, (i) we propose an influence estimation method that uses the Jacobian of the gradient of the generator's loss with respect to the discriminator's parameters (and vice versa) to trace how the absence of an instance in the discriminator's training affects the generator's parameters, and (ii) we propose a novel evaluation scheme, in which we assess harmfulness of each training instance on the basis of how GAN evaluation metric (e.g., inception score) is expected to change due to the removal of the instance. We experimentally verified that our influence estimation method correctly inferred the changes in GAN evaluation metrics. We also demonstrated that the removal of the identified harmful instances effectively improved the model's generative performance with respect to various GAN evaluation metrics.

## 1 INTRODUCTION

Generative adversarial networks (GANs) proposed by Goodfellow et al. (2014) are a powerful subclass of generative model, which is successfully applied to a number of image generation tasks (Antoniou et al., 2017; Ledig et al., 2017; Wu et al., 2016). The expansion of the applications of GANs makes improvements in the generative performance of models increasingly crucial.

An effective approach for improving machine learning models is to identify training instances that harm the model performance. Traditionally, statisticians manually screen a dataset for harmful instances, which misguide a model into producing biased predictions. Recent *influence estimation* methods (Khanna et al., 2019; Hara et al., 2019) automated the screening of datasets for deep learning settings, in which the sizes of both datasets and data dimensions are too large for users to manually determine the harmful instances. Influence estimation measures the effect of removing an individual training instance on a model's prediction without the computationally prohibitive cost of model retraining. The recent studies identified harmful instances by estimating how the loss value changes if each training instance is removed from the dataset.

Although previous studies have succeeded in identifying the harmful instances in supervised settings, the extension of their approaches to GAN is non-trivial. Previous approaches require that (i) the existence or absence of a training instance directly affects a loss value, and that (ii) the decrease in the loss value represents the harmfulness of the removed training instance. In GAN training, however, neither of the requirements is satisfied. (i) As training instances are only fed into the discriminator, they only indirectly affect the generator's loss, and (ii) the changes in the losses of GAN

do not necessarily capture how the removed instances harm the generative performance. This is because the ability of the loss to evaluate the generator is highly dependent on the performance of the discriminator.

To this end, (i) we propose an influence estimation method that uses the Jacobian of the gradient of the discriminator's loss with respect to the generator's parameters (and vice versa), which traces how the absence of an instance in the discriminator's training affects the generator's parameters. In addition, (ii) we propose a novel evaluation scheme to judge if an instance is harmful or not on the basis of *influence on GAN evaluation metric*, that is, how a GAN evaluation metric (e.g., inception score (Salimans et al., 2016)) changes if a given training instance is removed from the dataset. We identify harmful instances by estimating the influence on GAN evaluation metric by leveraging our influence estimation method.

We verified that the proposed influence estimation method correctly estimated the influence on GAN evaluation metrics across different settings of the dataset, model architecture, and GAN evaluation metrics. We also demonstrated that removing harmful instances, which were identified by the proposed method, effectively improved various GAN evaluation metrics.[1]

Our contributions are summarized as follows:

- We propose an influence estimation method that uses the Jacobian of the gradient of the discriminator's loss with respect to the generator's parameters (and vice versa), which traces how the absence of an instance in the discriminator's training affects the generator's parameters.

- We propose a novel evaluation scheme to judge if an instance is harmful or not on the basis of influence on GAN evaluation metrics rather than that on the loss value, and to leverage the proposed influence estimation method to identify harmful instances.

- We experimentally verified that our influence estimation method correctly inferred the influence on GAN evaluation metrics. Further, we demonstrated that the removal of the harmful instances suggested by the proposed method effectively improved the generative performance with respect to various GAN evaluation metrics.

## 2 PRELIMINARIES

**Notation**    For column vectors $\boldsymbol{a}, \boldsymbol{b} \in \mathbb{R}^p$, we denote the inner product by $\langle \boldsymbol{a}, \boldsymbol{b} \rangle = \sum_{i=1}^{p} a_i b_i$. For a function $f(\boldsymbol{a})$, we denote its gradient with respect to $\boldsymbol{a}$ by $\nabla_{\boldsymbol{a}} f(\boldsymbol{a})$. We denote the identity matrix of size $p$ by $\boldsymbol{I}_p$, the zero vector of length $p$ by $\boldsymbol{0}_p$, and the ones vector of length $p$ by $\boldsymbol{1}_p$.

**Generative Adversarial Networks (GAN)**    For simplicity, we consider an unconditional GAN that consists of the generator $G : \mathbb{R}^{d_{\boldsymbol{z}}} \to \mathbb{R}^{d_{\boldsymbol{x}}}$ and the discriminator $D : \mathbb{R}^{d_{\boldsymbol{x}}} \to \mathbb{R}$, where $d_{\boldsymbol{z}}$ and $d_{\boldsymbol{x}}$ are the number of dimensions of latent variable $\boldsymbol{z} \sim p(\boldsymbol{z})$ and data point $\boldsymbol{x} \sim p(\boldsymbol{x})$, respectively. The parameters of generator $\boldsymbol{\theta}_G \in \mathbb{R}^{d_G}$ and discriminator $\boldsymbol{\theta}_D \in \mathbb{R}^{d_D}$ are learned though the adversarial training; $G$ tries to sample realistic data while $D$ tries to identify whether the data is real or generated.

**Formulation of GAN Objectives**    For the generality, we adopt the formulation of Gidel et al. (2019) in which $G$ and $D$ try to minimize $\mathscr{L}_G$ and $\mathscr{L}_D$, respectively, to obtain the following *Nash equilibrium* $(\boldsymbol{\theta}_G^*, \boldsymbol{\theta}_D^*)$:

$$\boldsymbol{\theta}_G^* \in \arg\min_{\boldsymbol{\theta}_G} \mathscr{L}_G \left( \boldsymbol{\theta}_G, \boldsymbol{\theta}_D^* \right) \ \text{ and } \ \boldsymbol{\theta}_D^* \in \arg\min_{\boldsymbol{\theta}_D} \mathscr{L}_D \left( \boldsymbol{\theta}_G^*, \boldsymbol{\theta}_D \right). \tag{1}$$

For the latter part of this paper, we use a coupled parameter vector $\boldsymbol{\theta} := (\boldsymbol{\theta}_G, \boldsymbol{\theta}_D)^\top \in \mathbb{R}^{d_{\boldsymbol{\theta}} = d_G + d_D}$ when we refer to the whole parameters of GAN.

In this paper, we assume that $\mathscr{L}_G$ and $\mathscr{L}_D$ have the following forms[2]:

$$\mathscr{L}_G \left( \boldsymbol{\theta} \right) := \mathbb{E}_{\boldsymbol{z} \sim p(\boldsymbol{z})} \left[ f_G \left( \boldsymbol{z}; \boldsymbol{\theta} \right) \right], \quad \mathscr{L}_D \left( \boldsymbol{\theta} \right) := \mathbb{E}_{\boldsymbol{z} \sim p(\boldsymbol{z})} \left[ f_D^{[\boldsymbol{z}]} \left( \boldsymbol{z}; \boldsymbol{\theta} \right) \right] + \mathbb{E}_{\boldsymbol{x} \sim p(\boldsymbol{x})} \left[ f_D^{[\boldsymbol{x}]} \left( \boldsymbol{x}; \boldsymbol{\theta} \right) \right]. \tag{2}$$

---

[1]Code is at `https://github.com/hitachi-rd-cv/influence-estimation-for-gans`

[2]This covers the common settings of GAN objectives: the non-zero-sum game proposed by Goodfellow et al. (2014), Wasserstein distance (Arjovsky et al., 2017), and the least squares loss (Mao et al., 2017).

We can recover the original minimax objective by taking $f_G\left(\boldsymbol{z};\boldsymbol{\theta}\right) = \log\left(1 - D_{\boldsymbol{\theta}_D}\left(G_{\boldsymbol{\theta}_G}\left(\boldsymbol{z}\right)\right)\right)$, $f_D^{[\boldsymbol{z}]} = -f_G$, and $f_D^{[\boldsymbol{x}]}\left(\boldsymbol{x};\boldsymbol{\theta}\right) = -\log D_{\boldsymbol{\theta}_D}\left(\boldsymbol{x}\right)$.

**Adversarial SGD (ASGD)**    To make our derivation easier to understand, we newly formulate the parameter update of a GAN trained by stochastic gradient descent, which we call adversarial SGD (*ASGD*). For simplicity, this paper considers simultaneous training, in which the generator and the discriminator are simultaneously updated at a single step. We denote the dataset by $\mathcal{D}_{\boldsymbol{x}} := \{\boldsymbol{x}_n \sim p(\boldsymbol{x})\}_{n=1}^N$, which consists of $N$ data points. Let $\mathcal{S}_t \subset \{1,\ldots,N\}$ be a set of sample indices at the $t$-th step. We assume that the mini-batch of the $t$-th step consists of instances $\{\boldsymbol{x}_i\}_{i\in\mathcal{S}_t}$ and a set of latent variables $\mathcal{Z}_t = \{\boldsymbol{z}_l^{[t]} \sim p(\boldsymbol{z})\}_{l=1}^{|\mathcal{S}_t|}$, which are sampled independently at each step $t$. We denote the mean of $\mathscr{L}_G$ and $\mathscr{L}_D$ across the mini-batch by $\overline{\mathscr{L}}_G(\mathcal{Z};\boldsymbol{\theta}) := \frac{1}{|\mathcal{Z}|}\sum_{\boldsymbol{z}\in\mathcal{Z}} f_G\left(\boldsymbol{z};\boldsymbol{\theta}\right)$ and $\overline{\mathscr{L}}_D(\mathcal{S},\mathcal{Z};\boldsymbol{\theta}) := \frac{1}{|\mathcal{Z}|}\left(\sum_{\boldsymbol{z}\in\mathcal{Z}} f_D^{[\boldsymbol{z}]}\left(\boldsymbol{z};\boldsymbol{\theta}\right) + \sum_{i\in\mathcal{S}} f_D^{[\boldsymbol{x}]}\left(\boldsymbol{x}_i;\boldsymbol{\theta}\right)\right)$, respectively. The $t$-th step of ASGD updates the coupled parameters by $\boldsymbol{\theta}^{[t+1]} = \boldsymbol{\theta}^{[t]} - \boldsymbol{B}_t g\left(\mathcal{S}_t, \mathcal{Z}_t; \boldsymbol{\theta}^{[t]}\right)$, where

$$\boldsymbol{B}_t := \begin{pmatrix} \eta_G^{[t]}\boldsymbol{I}_{d_G} & \boldsymbol{O} \\ \boldsymbol{O} & \eta_D^{[t]}\boldsymbol{I}_{d_D} \end{pmatrix} \in \mathbb{R}^{d_{\boldsymbol{\theta}}\times d_{\boldsymbol{\theta}}}, \quad g\left(\mathcal{S},\mathcal{Z};\boldsymbol{\theta}\right) := \begin{pmatrix} \nabla_{\boldsymbol{\theta}_G}\overline{\mathscr{L}}_G\left(\mathcal{Z};\boldsymbol{\theta}\right) \\ \nabla_{\boldsymbol{\theta}_D}\overline{\mathscr{L}}_D\left(\mathcal{S},\mathcal{Z};\boldsymbol{\theta}\right) \end{pmatrix} \in \mathbb{R}^{d_{\boldsymbol{\theta}}}. \quad (3)$$

$\eta_G^{[t]} \in \mathbb{R}^+$ and $\eta_D^{[t]} \in \mathbb{R}^+$ are the learning rates of the $t$-th step for $\boldsymbol{\theta}_G$ and $\boldsymbol{\theta}_D$, respectively.

## 3    PROPOSED METHOD

This section explains the two main contributions of our paper: the influence estimation method for GANs that predicts how the removal of a training instance changes the output of the generator and the discriminator (Section 3.1), and two important parts of our instance evaluation scheme, that are, the definition of influence on GAN evaluation metric and its estimation algorithm (Section 3.2).

### 3.1    INFLUENCE ESTIMATION FOR GAN

We refer to influence estimation as the estimation of changes in a model's output under a training instance's absence. As the model's output changes through the changes in the model's parameters, we start with the definition of ASGD-Influence, which represents the changes in parameters, and then formulate its estimator.

**ASGD-Influence**    ASGD-Influence is defined on the basis of the following *counterfactual ASGD*. Let $\theta_{-j}^{[t]}$ denote the parameters at $t$-th step trained without using $j$-th training instance. Counterfactual ASGD starts optimization from $\boldsymbol{\theta}_{-j}^{[1]} = \boldsymbol{\theta}^{[1]}$ and updates the parameters of the $t$-th step by $\boldsymbol{\theta}_{-j}^{[t+1]} = \boldsymbol{\theta}_{-j}^{[t]} - \boldsymbol{B}_t g\left(\mathcal{S}_t \setminus \{j\}, \mathcal{Z}_t; \boldsymbol{\theta}_{-j}^{[t]}\right)$. We define ASGD-Influence $\Delta\boldsymbol{\theta}_{-j}$ as the parameter difference between counterfactual ASGD and ASGD at the final step $t = T$, namely $\Delta\boldsymbol{\theta}_{-j} := \boldsymbol{\theta}_{-j}^{[T]} - \boldsymbol{\theta}^{[T]}$.

**Estimator of ASGD-Influence**    Our estimator uses an approximation of the mean of the gradient. Let $\left(\nabla_{\boldsymbol{\theta}_G}\overline{\mathscr{L}}_G(\mathcal{Z};\boldsymbol{\theta}), \nabla_{\boldsymbol{\theta}_D}\overline{\mathscr{L}}_D(\mathcal{S},\mathcal{Z};\boldsymbol{\theta})\right)^\top$ be the joint gradient vector of the mini-batch. We introduce the Jacobian of the joint gradient vector of the $t$-th mini-batch with respect to $\boldsymbol{\theta}$:

$$\boldsymbol{J}_t := \begin{pmatrix} \boldsymbol{J}_{GG}^{[t]} & \boldsymbol{J}_{GD}^{[t]} \\ \boldsymbol{J}_{DG}^{[t]} & \boldsymbol{J}_{DD}^{[t]} \end{pmatrix} = \begin{pmatrix} \nabla_{\boldsymbol{\theta}_G}^2\overline{\mathscr{L}}_G\left(\mathcal{Z}_t;\boldsymbol{\theta}^{[t]}\right) & \nabla_{\boldsymbol{\theta}_D}\nabla_{\boldsymbol{\theta}_G}\overline{\mathscr{L}}_G\left(\mathcal{Z}_t;\boldsymbol{\theta}^{[t]}\right) \\ \nabla_{\boldsymbol{\theta}_G}\nabla_{\boldsymbol{\theta}_D}\overline{\mathscr{L}}_D\left(\mathcal{S}_t,\mathcal{Z}_t;\boldsymbol{\theta}^{[t]}\right) & \nabla_{\boldsymbol{\theta}_D}^2\overline{\mathscr{L}}_D\left(\mathcal{S}_t,\mathcal{Z}_t;\boldsymbol{\theta}^{[t]}\right) \end{pmatrix}. \quad (4)$$

When we assume both $\mathscr{L}_G(\boldsymbol{\theta})$ and $\mathscr{L}_G(\boldsymbol{\theta})$ are second-order differentiable with respect to $\boldsymbol{\theta}$, the first-order Taylor approximation gives $g\left(\mathcal{S}_t, \mathcal{Z}_t; \boldsymbol{\theta}_{-j}^{[t]}\right) - g\left(\mathcal{S}_t, \mathcal{Z}_t; \boldsymbol{\theta}^{[t]}\right) \approx \boldsymbol{J}_t\left(\boldsymbol{\theta}_{-j}^{[t]} - \boldsymbol{\theta}^{[t]}\right)$. With this approximation, we have

$$\boldsymbol{\theta}_{-j}^{[t+1]} - \boldsymbol{\theta}^{[t+1]} = \left(\boldsymbol{\theta}_{-j}^{[t]} - \boldsymbol{\theta}^{[t]}\right) - \boldsymbol{B}_t\left(g\left(\mathcal{S}_t, \mathcal{Z}_t; \boldsymbol{\theta}_{-j}^{[t]}\right) - g\left(\mathcal{S}_t, \mathcal{Z}_t; \boldsymbol{\theta}^{[t]}\right)\right)$$
$$\approx \left(\boldsymbol{I}_{d_{\boldsymbol{\theta}}} - \boldsymbol{B}_t\boldsymbol{J}_t\right)\left(\boldsymbol{\theta}_{-j}^{[t]} - \boldsymbol{\theta}^{[t]}\right), \forall j \notin \mathcal{S}_t. \quad (5)$$

For simplicity, we first focus on 1-epoch ASGD in which each instance appears only once. Let $\pi(j)$ be the step where the $j$-th instance is used. Considering the absence of $\nabla_{\boldsymbol{\theta}_D} f_D^{[\boldsymbol{x}]}(\boldsymbol{x}_j; \boldsymbol{\theta}^{[\pi(j)]})$ in the $\pi(j)$-th step of counterfactual ASGD, we have $\boldsymbol{\theta}_{-j}^{[\pi(j)+1]} - \boldsymbol{\theta}^{[\pi(j)+1]} = \frac{\eta_D^{[\pi(j)]}}{|\mathcal{S}_{\pi(j)}|} \left( \boldsymbol{0}_{d_G}, \nabla_{\boldsymbol{\theta}_D} f_D^{[\boldsymbol{x}]}(\boldsymbol{x}_j; \boldsymbol{\theta}^{[\pi(j)]}) \right)^\top$. By denoting $\boldsymbol{Z}_t := \boldsymbol{I}_{d_{\boldsymbol{\theta}}} - \boldsymbol{B}_t \boldsymbol{J}_t$ and recursively applying the approximation (5), we obtain

$$\Delta \boldsymbol{\theta}_{-j} \approx \frac{\eta_D^{[\pi(j)]}}{|\mathcal{S}_{\pi(j)}|} \boldsymbol{Z}_{T-1} \boldsymbol{Z}_{T-2} \cdots \boldsymbol{Z}_{\pi(j)+1} \left( \begin{matrix} \boldsymbol{0}_{d_G} \\ \nabla_{\boldsymbol{\theta}_D} f_D^{[\boldsymbol{x}]} \left( \boldsymbol{x}_j; \boldsymbol{\theta}^{[\pi(j)]} \right) \end{matrix} \right). \tag{6}$$

For the practical situation of $K$-epoch ASGD, in which the $j$-th instance is sampled $K$ times at $t = \pi_1(j), \ldots, \pi_K(j)$, the estimator of the ASGD-Influence is given by

$$\Delta \hat{\boldsymbol{\theta}}_{-j} := \sum_{k=1}^{K} \left( \prod_{s=1}^{T-\pi_k(j)-1} \boldsymbol{Z}_{T-s} \right) \frac{\eta_D^{[\pi_k(j)]}}{|\mathcal{S}_{\pi_k(j)}|} \left( \begin{matrix} \boldsymbol{0}_{d_G} \\ \nabla_{\boldsymbol{\theta}_D} f_D^{[\boldsymbol{x}]} \left( \boldsymbol{x}_j; \boldsymbol{\theta}^{[\pi_k(j)]} \right) \end{matrix} \right). \tag{7}$$

**Linear Influence** To estimate the influence on outputs, we introduce *linear influence* $L_{-j}^{[T]}(\boldsymbol{u}) := \langle \boldsymbol{u}, \Delta \boldsymbol{\theta}_{-j} \rangle$ of a given query vector $\boldsymbol{u} \in \mathbb{R}^{d_{\boldsymbol{\theta}}}$. If we take $\boldsymbol{u} = \nabla_{\boldsymbol{\theta}} f_G \left( \boldsymbol{z}; \boldsymbol{\theta}^{[T]} \right)$, the linear influence approximates the influence on the generator's loss $L_{-j}^{[T]}(\boldsymbol{u}) \approx f_G \left( \boldsymbol{z}; \boldsymbol{\theta}_{-j}^{[T]} \right) - f_G \left( \boldsymbol{z}; \boldsymbol{\theta}^{[T]} \right)$.

Let $\left( \boldsymbol{u}_G^{[t]\top} \in \mathbb{R}^{d_G}, \boldsymbol{u}_D^{[t]\top} \in \mathbb{R}^{d_D} \right) := \boldsymbol{u}^\top \boldsymbol{Z}_{T-1} \boldsymbol{Z}_{T-2} \cdots \boldsymbol{Z}_{t+1}$. The linear influence of the $j$-th instance is approximated by the proposed estimator:

$$L_{-j}^{[T]}(\boldsymbol{u}) \approx \left\langle \boldsymbol{u}, \Delta \hat{\boldsymbol{\theta}}_{-j} \right\rangle = \sum_{k=1}^{K} \frac{\eta_D^{[\pi_k(j)]}}{|\mathcal{S}_{\pi_k(j)}|} \left\langle \boldsymbol{u}_D^{[\pi_k(j)]}, \nabla_{\boldsymbol{\theta}_D} f_D^{[\boldsymbol{x}]} \left( \boldsymbol{x}_j; \boldsymbol{\theta}^{[\pi_k(j)]} \right) \right\rangle. \tag{8}$$

The estimation algorithm consists of two phases; *training phase* performs $K$-epoch ASGD by storing information $\mathcal{A}^{[t]} \leftarrow (\mathcal{S}_t, \eta_G^{[t]}, \eta_D^{[t]}, \boldsymbol{\theta}^{[t]}, \mathcal{Z}_t)$ and *inference phase* calculates (8) using $\mathcal{A}^{[1]}, \ldots, \mathcal{A}^{[T-1]}$. See Appendix A for the detailed algorithm.

## 3.2 INFLUENCE ON GAN EVALUATION METRIC

This section explains our proposal of a new evaluation approach for data screening for GANs. Firstly we propose to evaluate harmfulness of an instance on the basis of influence on GAN evaluation metrics. Secondly we propose to leverage the influence-estimation algorithm explained in Section 3.1 to identify harmful instances with respect to the GAN evaluation metrics.

**Influence on GAN Evaluation Metric** Let $V(\mathcal{D})$ be a GAN evaluation metric that maps a set of data points $\mathcal{D} := \{\tilde{\boldsymbol{x}}_m \in \mathbb{R}^{d_{\boldsymbol{x}}}\}_{m=1}^{M}$ into a scalar value that gives the performance measure of $G$. Let generated dataset $\mathcal{D}_G(\mathcal{Z}; \boldsymbol{\theta}_G) := \{G(\boldsymbol{z}; \boldsymbol{\theta}_G) | \boldsymbol{z} \in \mathcal{Z}\}$. Using a set of latent variables $\mathcal{Z} := \{\tilde{\boldsymbol{z}}_m \sim p(\boldsymbol{z})\}_{m=1}^{M}$ that is sampled independently from the training, we define the influence on GAN evaluation metric by

$$\Delta V_{-j}^{[T]} := V \left( \mathcal{D}_G \left( \mathcal{Z}; \boldsymbol{\theta}_{G,-j}^{[T]} \right) \right) - V \left( \mathcal{D}_G \left( \mathcal{Z}; \boldsymbol{\theta}_G^{[T]} \right) \right), \tag{9}$$

where $\boldsymbol{\theta}_{G,-j}^{[T]}$ and $\boldsymbol{\theta}_G^{[T]}$ are the generator parameters of counterfactual ASGD and the ASGD of the $T$-th step, respectively.

**Estimation Algorithm** In order to build the estimation algorithm of the influence on GAN evaluation metric, we focus on an important property of some common evaluation metrics for which the gradient with respect to the element of their input $\nabla_{\tilde{\boldsymbol{x}}_m} V(\mathcal{D})$ is computable. For example, Monte Carlo estimation of inception score has a form of $\exp(\frac{1}{|\mathcal{D}|} \sum_{\tilde{\boldsymbol{x}}_m \in \mathcal{D}} \mathbb{KL}(p_c(y|\tilde{\boldsymbol{x}}_m) \| p_c(y)))$ where $p_c$ is a distribution of class label $y$ drawn by a pretrained classifier. When the classifier is trained using back-propagation, $\nabla_{\tilde{\boldsymbol{x}}_m} V(\mathcal{D})$ is computable.

Here, we assume $V(\mathcal{D})$ is first-order differentiable with respect to $\tilde{x}_m$. From the chain rule, we have a gradient of the GAN evaluation metrics with respect to $\boldsymbol{\theta}$:

$$\nabla_{\boldsymbol{\theta}} V(\mathcal{D}_G(\mathcal{Z}; \boldsymbol{\theta}_G^{[T]})) = \begin{pmatrix} \sum_{n=1}^{M} \nabla_{\boldsymbol{\theta}_G} \nabla_{\tilde{x}_n} V\left(\mathcal{D}_G\left(\mathcal{Z}; \boldsymbol{\theta}_G^{[T]}\right)\right) \\ \mathbf{0}_{d_D} \end{pmatrix}. \tag{10}$$

Our estimation algorithm performs the inference phase of linear influence taking $\boldsymbol{u} = \nabla_{\boldsymbol{\theta}} V(\mathcal{D}_G(\mathcal{Z}; \boldsymbol{\theta}_G^{[T]}))$ in order to obtain the approximation $L_{-j}^{[T]}(\nabla_{\boldsymbol{\theta}} V(\mathcal{D}_G(\mathcal{Z}; \boldsymbol{\theta}_G^{[T]}))) \approx \Delta V_{-j}^{[T]}$.

## 4 RELATED STUDIES

**SGD-Influence** Hara et al. (2019) proposed a novel definition of the influence called *SGD-Influence* and its estimator, which greatly inspired us to propose the influence estimation method for GANs. Suppose a machine learning model with parameters $\boldsymbol{\phi} \in \mathbb{R}^{d_\phi}$ is trained to minimize the mean of the loss $\frac{1}{N} \sum_{n=1}^{N} \mathscr{L}(\chi_n; \boldsymbol{\phi})$ across the training instances $\chi_1, \ldots, \chi_N$. Let the mean of the loss of the mini-batch $\overline{\mathscr{L}}(\mathcal{S}; \boldsymbol{\phi}) := \frac{1}{|\mathcal{S}|} \sum_{i \in \mathcal{S}} \mathscr{L}(\chi_i; \boldsymbol{\phi})$. They introduced two SGD steps with learning rate $\eta_t \in \mathbb{R}^+$: SGD given by $\boldsymbol{\phi}^{[t+1]} = \boldsymbol{\phi}^{[t]} - \eta_t \nabla_{\boldsymbol{\phi}} \overline{\mathscr{L}}\left(\mathcal{S}_t; \boldsymbol{\phi}^{[t]}\right)$, and counterfactual SGD given by $\boldsymbol{\phi}_{-j}^{[t+1]} = \boldsymbol{\phi}_{-j}^{[t]} - \eta_t \nabla_{\boldsymbol{\phi}} \overline{\mathscr{L}}\left(\mathcal{S}_t \setminus \{j\}; \boldsymbol{\phi}_{-j}^{[t]}\right)$. Their estimator of SGD-Influence $\boldsymbol{\phi}_{-j}^{[T]} - \boldsymbol{\phi}^{[T]}$ is based on the following approximation:

$$\boldsymbol{\phi}_{-j}^{[t+1]} - \boldsymbol{\phi}^{[t+1]} \approx \left(\boldsymbol{I}_{d_\phi} - \eta_t \nabla_{\boldsymbol{\phi}}^2 \overline{\mathscr{L}}\left(\mathcal{S}_t; \boldsymbol{\phi}^{[t]}\right)\right)\left(\boldsymbol{\phi}_{-j}^{[t]} - \boldsymbol{\phi}^{[t]}\right) , \forall j \notin \mathcal{S}_t. \tag{11}$$

Hara et al. (2019) also identified harmful instances for classification based on linear influence of the cross-entropy loss estimated using a validation dataset. Removing the estimated harmful instances with their approach demonstrated improvements in the classification accuracy.

Our approach differs from Hara et al. (2019)'s work in two ways. Firstly, our approach uses the Jacobian of the joint gradient vector $\boldsymbol{J}_t$ instead of the Hessian of the mean loss $\nabla_{\boldsymbol{\phi}}^2 \overline{\mathscr{L}}\left(\mathcal{S}_t; \boldsymbol{\phi}^{[t]}\right)$. As long as $\mathscr{L}_G \neq \mathscr{L}_D$, $\boldsymbol{J}_t$ is asymmetric and inherently different from the Hessian. Moreover, a source of the asymmetry $\boldsymbol{J}_{GD}^{[t]}$ plays an important role in transferring the effect of removal of a training instance from the discriminator to the generator. Let $\boldsymbol{\theta}_{G,-j}^{[t]} - \boldsymbol{\theta}_G^{[t]} \in \mathbb{R}^{d_G}$ and $\boldsymbol{\theta}_{D,-j}^{[t]} - \boldsymbol{\theta}_D^{[t]} \in \mathbb{R}^{d_D}$ be ASGD-Influence on $\boldsymbol{\theta}_G$ and $\boldsymbol{\theta}_D$ of the $t$-th step, respectively. The upper blocks of (5) can be rewritten as

$$\boldsymbol{\theta}_{G,-j}^{[t+1]} - \boldsymbol{\theta}_G^{[t+1]} \approx \left(\boldsymbol{I}_{d_D} - \eta_G^{[t]} \boldsymbol{J}_{GG}^{[t]}\right)\left(\boldsymbol{\theta}_{G,-j}^{[t]} - \boldsymbol{\theta}_G^{[t]}\right) + \eta_G^{[t]} \boldsymbol{J}_{GD}^{[t]}\left(\boldsymbol{\theta}_{D,-j}^{[t]} - \boldsymbol{\theta}_D^{[t]}\right). \tag{12}$$

Note that $\boldsymbol{J}_{GD}^{[t]}$ transfers the $t$-th step of ASGD-Influence on $\boldsymbol{\theta}_D$ to the next step of ASGD-Influence on $\boldsymbol{\theta}_G$. The Hessian of Hara et al. (2019), which uses a single combination of the parameters and the loss function, cannot handle this transfer between the two models. Secondly, we use the influence on GAN evaluation metrics for identifying harmful instances rather than that on the loss value. This alleviates the problem of the GAN's loss not representing the generative performance.

**Influence Function** Koh & Liang (2017) proposed influence estimation method that incorporated the idea of influence function (Cook & Weisberg, 1980) in robust statistics. They showed that influences on parameters and predictions can be estimated with the influence function assuming the satisfaction of the optimality condition and strong convexity of the loss function. They also identified harmful instances on the basis of the influence on the loss value, assuming consistency of the loss value with the task performance.

Our influence estimation method is designed to eliminate these assumptions because normally GAN training does not satisfy the assumptions regarding the optimality condition, the convexity in the loss function, and the consistency of the loss value with the performance.

## 5 EXPERIMENTS

We evaluated the effectiveness of the proposed method in two aspects: the accuracy of influence estimation on GAN evaluation metrics (Section 5.1), and the improvement in generative performance by removing estimated harmful instances (Section 5.2)

**GAN Evaluation Metrics**   In both experiments, we used three GAN evaluation metrics: average log-likelihood (ALL), inception score (IS), and Fréchet inception distance (FID) (Heusel et al., 2017). ALL is the de-facto standard for evaluating generative models (Tolstikhin et al., 2017). Let $\mathcal{Z}' := \{z_n' \sim p(z)\}_{n=1}^{N'}$ and $\mathcal{D}_x' := \{x_n' \sim p(x)\}_{n=1}^{N'}$, which is sampled separately from $p(z)$ and the training dataset $\mathcal{D}_x$, respectively. ALL measures the likelihood of the true data under the distribution that is estimated from generated data using kernel density estimation. We calculated ALL of $\mathcal{D}_x'$ under the distribution estimated from generated dataset $\mathcal{D}_G(\mathcal{Z}'; \theta_G^{[T]})$. Recall $\mathcal{Z}'$ is the set of latent variables sampled independently from the training (Section 3.2). FID measures Fréchet distance between two sets of feature vectors of real images $\mathcal{D}_x'$ and those of generated images $\mathcal{D}_G(\mathcal{Z}'; \theta_G^{[T]})$. The feature vectors are calculated on the basis of a pre-trained classifier. Larger values of ALL and IS and a smaller value of FID indicate the better generative performance. See Appendix C.1 for the detailed setting of each GAN evaluation metric.

## 5.1   EXPERIMENT 1: ESTIMATION ACCURACY

We ran the influence estimation method on GANs to estimate influence on various GAN evaluation metrics, and then compared the estimated influence with true influence. The detailed setup can be found in Appendix C.2.

**Setup**   ALL is known to be effective for low-dimensional data distributions (Borji, 2019) and both FID and IS are effective for image distributions. We thus prepared two different setups: fully-connected GAN (FCGAN) trained with 2D multivariate normal distribution (2D-Normal) for ALL, and DCGAN (Radford et al., 2016) trained with MNIST (LeCun et al., 1998) for IS and FID. IS and FID require classifiers to obtain class label distribution and feature vectors, respectively. We thus trained CNN classifier of MNIST[3] using $\mathcal{D}_x'$. We set $N = 10\text{k}$ and $N' = |\mathcal{D}_x'| = |\mathcal{Z}'| = 10\text{k}$.

The experiment was conducted as follows. Firstly, we ran the $K$-epoch of the training phase of linear influence with the training dataset $\mathcal{D}_x$. We determined $K = 50$ since we observed the convergence of GAN evaluation metrics at $K = 50$. For IS and FID, we trained the classifier using $\mathcal{D}_x'$ and corresponding labels. We then randomly selected 200 target instances from $\mathcal{D}_x$. We obtained estimated influence on GAN evaluation metrics of each target instance by performing the inference phase of linear influence with $u = \nabla_\theta V(\mathcal{D}_G(\mathcal{Z}'; \theta_G^{[T]}))$. The true influence of each target instance was computed by running the counterfactual ASGD.

We used the same evaluation measures as the previous work (Hara et al., 2019): Kendall's Tau and the Jaccard index. Kendall's Tau measures the ordinal correlation between the estimated and true influence on GAN evaluation metrics. It has a value of 1 when the orders of the two sets of values are identical. For the Jaccard index, we selected 10 instances with the largest positive and largest negative influence values to construct a set of 20 critical instances. The Jaccard index is equal to 1 when a set of estimated critical instances is identical to that of true critical instances.

To investigate the relationship between a number of tracing back steps and the estimation accuracy, we also evaluated the influence on GAN evaluation metrics of $k$-epoch ASGD. In $k$-epoch training, both inference phase of linear influence and the counterfactual ASGD traced back only $k \leq K$ epochs from the latest epoch $K$. We varied $k = 1, 5, 10, 20, 50$ and ran the experiment ten times for each $k$ by changing the random seeds of the experiments.

**Results**   Figure 1 shows the average Kendal's Tau and the Jaccard index of the repeated experiments. Hereinafter, we use $p < .05$ to judge the statistical significance of the results. For all $k$, Kendall's Tau and the Jaccard index of estimated influence on ALL were statistically significantly better than the result in which the order of estimated influence values were random (random case). Even in the more difficult setups of IS and FID, which handled the high-dimensional dataset and complex architecture, the results were statistically significantly better than that of the random case except for Jaccard index of IS with $k = 50$. We also observed the estimation accuracy dropped as $k$ increased. This reflects the nature of our estimator that recursively performs linear approximation

---

[3]Although the original IS and FID use Inception Net (Szegedy et al., 2016) trained with ImageNet, we instead adopted a domain-specific classifier as encouraged by several studies (Zhou et al., 2018; Liu et al., 2018) to alleviate the domain mismatch with ImageNet.

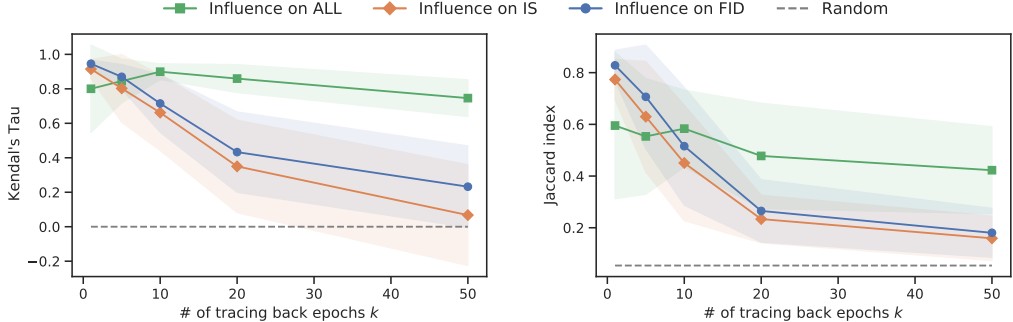

Figure 1: Average Kendall's Tau (±std) (left) and the Jaccard index (±std) (right) calculated from true and estimated influence on ALL, IS, and FID.

as many times as the number of steps. We thus conclude that when the required number of tracing back steps is small enough, our influence estimation method is effective and the estimated influence on GAN evaluation metric is useful for identifying harmful instances.

## 5.2 EXPERIMENT 2: DATA CLEANSING

We investigated if removing identified harmful instances actually improved the generative performance to evaluate the effectiveness of our proposed method for *data cleansing*. We define data cleansing as an attempt to improve GAN evaluation metrics by removing a set of training instances. See appendix C.3 for the detailed settings.

**Setup** We studied the data cleansing for the two setups explained in the previous section: 2D-Normal with FCGAN and MNIST with DCGAN. We mostly followed the settings of Section 5.1 but set training dataset size $N = 50$k for both setups.

We identified harmful instances in 2D-Normal training dataset using estimated influence on ALL, and those in MNIST using estimated influence on IS and FID. We considered a training instance was harmful when it had negative (positive) influence on FID (ALL or IS).

For both setups, we also selected instances using baseline approaches: anomaly detection method, influence on the discriminator loss, and random values. For anomaly detection, we adopted isolation forest (Liu et al., 2008). Isolation forest fitted the model using the data points of $\mathcal{D}_{\boldsymbol{x}}$ for 2D-Normal and feature vectors of the classifier of $\mathcal{D}_{\boldsymbol{x}}$ for MNIST. We adopted the selection based on the influence on the discriminator loss to verify our assumption that the influence on the loss does not represent the harmfulness of the instances. Influence on the discriminator loss was calculated on the expected loss of $\mathscr{L}_D(\boldsymbol{\theta})$ with $\mathcal{D}_G(\mathcal{Z}'; \boldsymbol{\theta}_G^{[T]})$ and $\mathcal{D}'_{\boldsymbol{x}}$. We considered instances with negative influence were harmful.

We conducted the experiments as follows. After the training phase of $K$ epoch, we determined $n_h < N$ harmful instances with the proposed approach and baselines. Then, we ran counterfactual ASGD with the determined harmful instances excluded. For the reliable estimation accuracy of influence and reasonable costs of the computation and storage, the inference phase traced back only 1-epoch from the last epoch, and counterfactual ASGD only re-ran the latest epoch. We tested with various $n_h$.

We refer to the generator of the final model as the cleansed generator and denote its parameters by $\boldsymbol{\theta}_G^\star$. We evaluated the cleansed generator with test GAN evaluation metrics $V(\mathcal{D}_G(\mathcal{Z}_{test}); \boldsymbol{\theta}_G^\star)$, in which a set of test latent variables $\mathcal{Z}_{test}$ was obtained by sampling $N_{test}$ times from $p(\boldsymbol{z})$ independently from $\mathcal{Z}'$ and $\mathcal{Z}_1, \ldots, \mathcal{Z}_T$. Test ALL and FID used a test dataset $\mathcal{D}_{test} := \{\boldsymbol{x}_{test}^{[n]} \sim p(\boldsymbol{x})\}_{n=1}^{N_{test}}$ that consists of instances newly sampled from 2D-Normal and instances in the original test dataset of MNIST, respectively. We set $N_{test} = 10$k and ran the experiment 15 times with different random seeds.

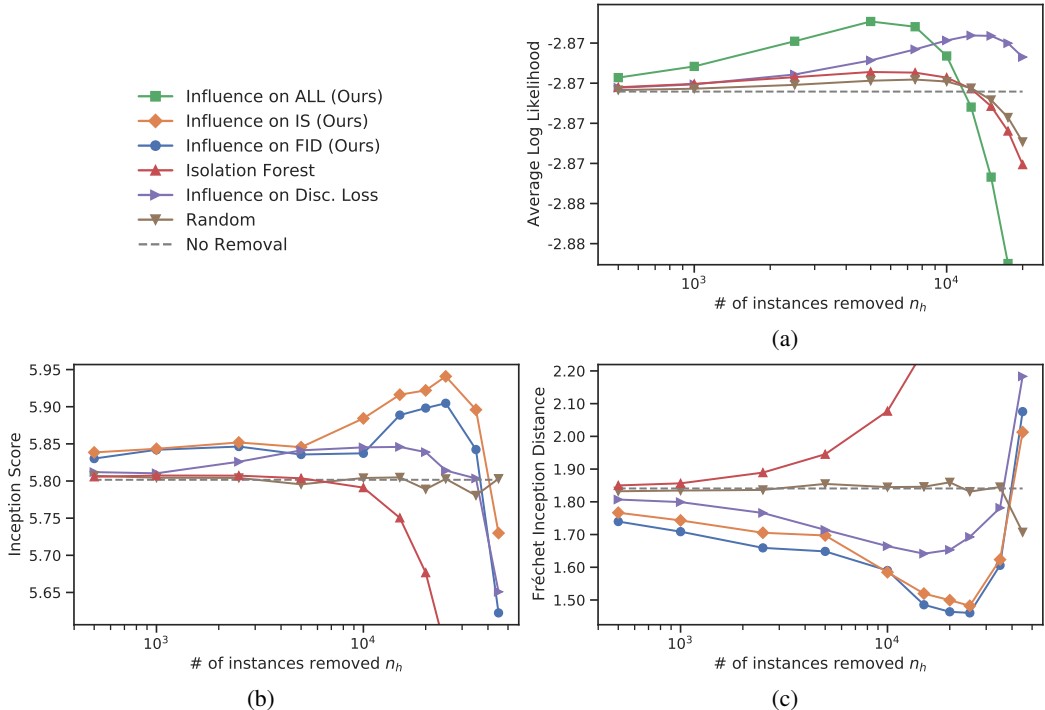

Figure 2: Average test ALL (a), IS (b), and FID (c) after the data cleansing. Larger values in (a) and (b), a smaller value in (c) indicate the better generative performance. Error bars and plots of too large or small values are omitted for better visibility. See Appendix C.3 for full results.

**Quantitative Results**    Figure 2 shows the average test GAN evaluation metrics of the repeated experiments for each selection approach. For the data cleansing on 2D-Normal, the proposed approach with influence on ALL showed statistically significant improvement from the original model and it outperformed the baselines (Figure 2a). For the MNIST setup, our approach with influence on FID and IS statistically significantly improved FID (Figure 2c) and IS (Figure 2b), respectively. They also outperformed the baselines. In addition, the results indicate that data cleansing based on the influence on a specific GAN evaluation metric is also effective for another metric that is not used for the selection; removing harmful instances based on the influence on FID (IS) statistically significantly improved IS (FID). However, we make no claim that the proposed method can improve all the other evaluation metrics, such as Kullback-Leibler divergence. This is because all the current GAN evaluation metrics have their own weaknesses (e.g., IS fails to detect whether a model is trapped into one bad mode (Zhou et al., 2018)), and the proposed method based on those GAN evaluation metrics cannot inherently avoid their weaknesses. These improvements thus can be observed only in a subclass of GAN evaluation metrics. Further evaluation of data cleansing with our method should incorporate the future improvements of the GAN evaluation metrics.

While the improvements were smaller than the proposed approach, we also observed that data cleansing based on the influence on the discriminator loss improved all the GAN evaluation metrics. This counter-intuitive result indicates that the discriminator loss weakly measures the performance of the generator that is trained along with the discriminator.

**Qualitative Results**    We examined the characteristics of instances that were evaluated to be harmful by our method. Overall, we observed that our method tends to judge instances as harmful when they belong to regions from which the generators sample too frequently compared to the true distribution. Figure 3 shows the estimated harmfulness of the training instances of 2D-Normal and the distribution of the generated samples. The proposed approach with influence on ALL evaluated the instances around lower-left and upper-right regions to be harmful (Figure 3a). These regions correspond to the regions where the generated distribution has higher density than that of the true distribution (Figure 3b "No removal" and "True"). Similar characteristics were seen in harmful

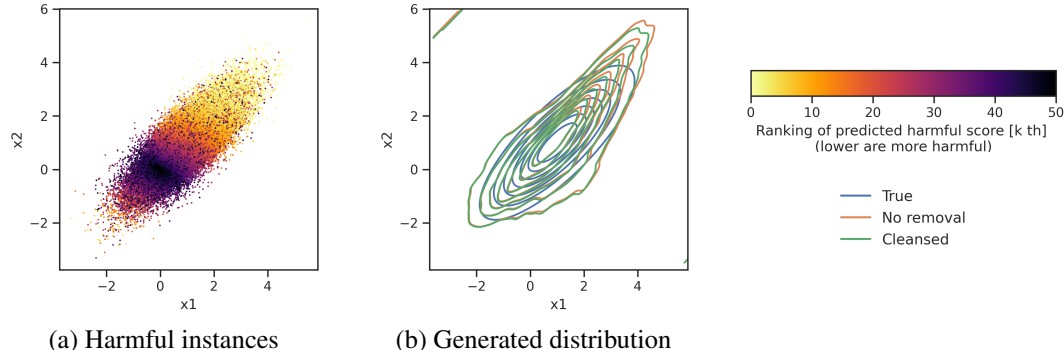

(a) Harmful instances      (b) Generated distribution

Figure 3: Harmfulness of 2D-Normal instances suggested using influence on ALL (a) and changes in the generator's distribution (b). (b) includes plots of the true distribution (True) and generator's distributions before (No removal) and after (Cleansed) the data cleansing with $n_h = 5.0$k.

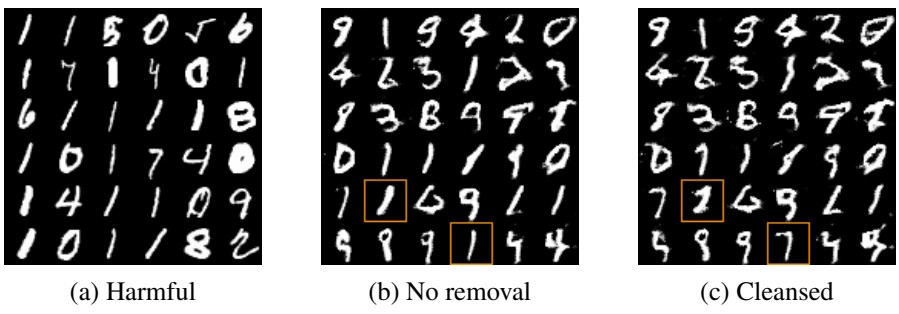

(a) Harmful      (b) No removal      (c) Cleansed

Figure 4: Top 36 harmful MNIST instances predicted on the basis of influence on FID (a), and the test generated samples before (b) and after (c) the data cleansing with $n_h = 25.0$k. (a) and (b) use the same series of test latent variables in $\mathcal{Z}_{test}$.

MNIST instances suggested by our approach with influence on FID. A large number of samples from class 1 were regarded as harmful as shown in Figure 4a, when the generator sampled images of the digit 1 too frequently (Figure 4b).

We also investigated how the data cleansing by our approach visually changed the generated samples. As seen from the distributions in Figure 3b, the probability density in the upper-right region decreased after the data cleansing (from "No removal" to "Cleansed"). As a result, the generator distribution moved closer to the true distribution. The same effect was observed in a visually more interesting form in the data cleansing for MNIST. The generated samples originating from some latent variables changed from the image of digit 1 to that of other digits after the data cleansing based on the estimated influence on FID (highlighted samples in Figure 4c). We suppose this effect improved the diversity in the generated samples, resulting in better FID and IS.

## 6 CONCLUSION

We proposed an influence estimation method for GAN that uses the Jacobian of the gradient of the discriminator's loss with respect to the generator's parameters (and vice versa), which traces how the absence of an instance in the discriminator's training affects the generator's parameters. We also proposed a novel evaluation scheme to judge if an instance is harmful or not on the basis of the influence on GAN evaluation metrics rather than that on the loss value, and to leverage the proposed influence estimation method to identify harmful instances. We experimentally verified that estimated and true influence on GAN evaluation metrics had a statistically significant correlation. We also demonstrated removing identified harmful instances effectively improved the generative performance with respect to various GAN evaluation metrics.

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

---

**Algorithm 1** Training Phase

Initialize the parameter $\boldsymbol{\theta}^{[1]}$
Initialize the sequence as null: $\mathcal{A} \leftarrow \emptyset$
**for** $t = 1, 2, \ldots, T-1$ **do**
 *// sample latent variables*
 $\mathcal{Z}_t = \{ \boldsymbol{z}_l^{[t]} \sim p(\boldsymbol{z}) \}_{l=1}^{|\mathcal{S}_t|}$
 *// store information*
 $\mathcal{A}^{[t]} \leftarrow \left( \mathcal{S}_t, \eta_G^{[t]}, \eta_D^{[t]}, \boldsymbol{\theta}^{[t]}, \mathcal{Z}_t \right)$
 *// update parameters*
 $\boldsymbol{\theta}^{[t+1]} = \boldsymbol{\theta}^{[t]} - \boldsymbol{B}_t g \left( \mathcal{S}_t, \mathcal{Z}_t; \boldsymbol{\theta}^{[t]} \right)$
**end for**

---

**Algorithm 2** Inference Phase

**Require:** $\boldsymbol{u} = \left( \boldsymbol{u}_G \in \mathbb{R}^{d_G}, \ \boldsymbol{u}_D \in \mathbb{R}^{d_D} \right)^\top$
Initialize the influence: $L_{-j}^{[T]}(\boldsymbol{u}) \leftarrow 0$
**for** $t = T-1, T-2, \ldots, 1$ **do**
 *// load information*
 $\left( \mathcal{S}_t, \eta_G^{[t]}, \eta_D^{[t]}, \boldsymbol{\theta}^{[t]}, \mathcal{Z}_t \right) \leftarrow \mathcal{A}^{[t]}$
 *// update the linear influence of jth instance*
 **if** $j \in \mathcal{S}_t$ **then**
  $L_{-j}^{[T]}(\boldsymbol{u}) \mathrel{+}= \frac{\eta_D^{[t]}}{|\mathcal{S}_t|} \left\langle \boldsymbol{u}_D, \nabla_{\boldsymbol{\theta}_D} f_D^{[\boldsymbol{x}]}(\boldsymbol{x}_j; \boldsymbol{\theta}^{[t]}) \right\rangle$
 **end if**
 *// update* $\boldsymbol{u}$
 $\boldsymbol{u} \mathrel{-}= \boldsymbol{u}^\top \boldsymbol{B}_t \boldsymbol{J}_t$
**end for**

---

## A  ALGORITHM FOR LINEAR INFLUENCE

The proposed estimation algorithm for linear influence, which is explained in Section 3.1, is divided into the training phase (Algorithm 1) and inference phase (Algorithm 2).

The training phase executes ASGD training while storing the mini-batch indices $\mathcal{S}_t$, the learning rate $\eta_G^{[t]}, \eta_D^{[t]}$, the parameters $\boldsymbol{\theta}^{[t]}$ and the sampled latent variable $\mathcal{Z}_t$ into the information $\mathcal{A}^{[t]}$ at each step.

In the inference phase, $L_{-j}^{[T]}(\boldsymbol{u})$ is estimated by the recursive calculation. First, we set $L_{-j}^{[T]}(\boldsymbol{u})$ to 0 and set the query vector $\boldsymbol{u}$. The information $\mathcal{A}^{[t]}$, which is obtained in the training phase, is read in the order of $t = T-1, T-2, \ldots, 1$. When $j \in \mathcal{S}_t$, $L_{-j}^{[T]}(\boldsymbol{u})$ is updated using (8). Let $\boldsymbol{u}_t = \left( \boldsymbol{u}_G^{[t]}, \boldsymbol{u}_D^{[t]} \right)^\top$. Each step updates $\boldsymbol{u}$ based on $\boldsymbol{u}_{t+1} = \boldsymbol{u}_t^\top \boldsymbol{Z}_t = \boldsymbol{u}_t^\top (\boldsymbol{I}_{d_{\boldsymbol{\theta}}} - \boldsymbol{B}_t \boldsymbol{J}_t)$. A naive calculation of $\boldsymbol{u}_t^\top \boldsymbol{J}_t$ requires $O\left(d_{\boldsymbol{\theta}}^2\right)$ memory to store the matrix $\boldsymbol{J}_t$, which can be prohibitive for very large models. We can avoid this difficulty by directly computing $\boldsymbol{u}_t^\top \boldsymbol{J}_t$ without the explicit computation of $\boldsymbol{J}_t$. Because $\boldsymbol{u}_t^\top \boldsymbol{J}_t = \nabla_{\boldsymbol{\theta}} \left\langle \boldsymbol{u}_t, (\nabla_{\boldsymbol{\theta}_G} \mathscr{L}_G, \nabla_{\boldsymbol{\theta}_D} \mathscr{L}_D)^\top \right\rangle$, we need only to compute the derivative of the inner product of $\boldsymbol{u}_t$ and the joint gradient vector.

Our algorithm also covers the alternating gradient descent, in which the two models alternatively update their parameters at each step. By taking $\eta_G^{[t]}$ and $\eta_D^{[t]}$ such that they alternatively take 0 at each step, we can have ASGD and the estimator of ASGD-Influence for the alternating gradient descent. The implementation of linear influence for the alternating gradient descent is available in our repository[4].

## B  OTHER RELATED WORKS

**Anomaly Detection** A typical approach for identifying harmful instances is outlier detection. Outlier detection is used to remove abnormal instances from the training set before training the model to ensure that the model is not affected by the abnormal instances. For tabular data, there are several popular methods, such as One-class support vector machine (Schölkopf et al., 2001), local outlier factor (Breunig et al., 2000), and isolation forest (Liu et al., 2008). Although these methods can find abnormal instances, they are not necessarily harmful for the resulting models, as we showed in the experiment.

**Training GAN from Noisy Images** One typical type of data that harm generative performance is noisy images. AmbientGAN (Bora et al., 2018) and noise-robust GAN (Kaneko & Harada, 2020)

---

[4]https://github.com/hitachi-rd-cv/influence-estimation-for-gans

Table 1: Model architecture of CNN classifier of MNIST in Section 5.1 and 5.2.

| Stage | Operation | Stride | Filter Shape | Bias | Norm. | Activation | Output |
|-------|-----------|--------|--------------|------|-------|------------|--------|
| 0 | Input | - | - | - | - | - | [28, 28, 1] |
| 1 | Conv2D | 1 | [5, 5] | ✓ | - | Sigmoid | [25, 25, 8] |
| 2 | Conv2D | 1 | [5, 5] | ✓ | - | Sigmoid | [12, 12, 8] |
| 3 | MaxPooling | 2 | [2, 2] | - | - | Sigmoid | [392] |
| 4 | Linear | 1 | - | ✓ | - | Sigmoid | [128] |
| 5 | Linear | 1 | - | ✓ | - | Sigmoid | [10] |

are learning algorithms that make it possible to train a clean image generator from noisy images. The difference between these studies and ours is that these studies assume that the noise (e.g., Gaussian noise on pixels) given independently from the data distribution of the clean images is the only problem. However, some instances can affect the performance even if the instances are drawn only from the data distribution, which is the case robust statistics (Huber, 2004) typically focuses on. Our experiment 5.2 indicates that the model performance depends not only on noisy images but also on a non-negligible number of harmful instances in the original dataset.

# C   DETAILED EXPERIMENTAL SETTINGS AND RESULTS

## C.1   GAN EVALUATION METRICS

We adopted Gaussian kernel with the band-width 1 for kernel density estimation used in ALL. The architecture of CNN classifier of MNIST used for IS and FID can be found in Table 1. We selected the output of the 4th layer for the feature vectors for FID.

## C.2   EXPERIMENT 1: ESTIMATION ACCURACY

**Setup**   In the experiment of Section 5.1, we adopted the hyper parameters shown in Table 2. We trained fullly-connected GAN (FCGAN) for 2D multivariate normal distribution, in which the both $G$ and $D$ has 1 hidden layer of $h_G$ and $h_D$ units, respectively (Table 3). 2D-Normal is given by $\mathcal{N}(\boldsymbol{\mu}, \boldsymbol{\Sigma})$, in which the mean vector $\boldsymbol{\mu} = \mathbf{1}_2$ and the covariance matrix $\boldsymbol{\Sigma} = ((1, 0.8), (0.8, 1))^\top$. DCGAN consists of transposed convolution (or deconvolution) layers and convolution layers (Table 4). The channels of the both layers in $G$ and $D$ were determined by $h_G$ and $h_D$, respectively. We used Layer Normalization (Ba et al., 2016) for the layers shown in Table 4 for the stability of the training. We also introduced the L2-norm regularization with the rate $\gamma \in \mathbb{R}^+$ for all the kernels of both FCGAN and DCGAN. We used the non-zero-sum game objective of the original paper (Goodfellow et al., 2014) in which $G$ tries to minimize $-D_{\boldsymbol{\theta}_D}\left(G_{\boldsymbol{\theta}_G}\left(\boldsymbol{z}\right)\right)$ for both models.

## C.3   EXPERIMENT 2: DATA CLEANSING

**Setup**   We adopted the same architecture as the Section 5.1 (Table 3) for FCGAN and slightly different architecture (Table 4) in which $h_G$ and $h_D$ are larger (Table 5) for DCGAN. Other hyper parameters followed Table 5. We also provide visual explanations of the data settings in the experiments with influence on ALL, IS, and FID in Figure 5, 6, and 7, respectively.

**Results**   Table 6-8 show the detailed results of Figure 2. And they clarify with which $n_h$ and selection approach the test GAN evaluation metrics were statistically significantly improved.

Table 2: Hyper parameters in Section 5.1.

| | $K$ | $\eta_G^{[t]}$ | $\eta_D^{[t]}$ | $N$ | $N'$ | $\mathcal{S}_t$ | $\gamma$ | $h_G$ | $h_D$ |
|---|---|---|---|---|---|---|---|---|---|
| 2D-Normal | 50 | $10^{-3}$ | $10^{-3}$ | 10k | 10k | 100 | $10^{-3}$ | 32 | 64 |
| MNIST | 50 | $10^{-3}$ | $10^{-3}$ | 10k | 10k | 100 | $10^{-3}$ | 8 | 8 |

Table 3: Model Architecture of FCGAN in Section 5.1 and 5.2.

| Net. | Stage | Operation | Bias | Activation | Output |
|---|---|---|---|---|---|
| - | 0 | Input | - | - | [10] |
| $G$ | 1 | Linear | ✓ | ReLU | [$h_G$] |
| $G$ | 2 | Linear | ✓ | Tanh | [2] |
| $D$ | 3 | Linear | ✓ | ReLU | [$h_D$] |
| $D$ | 4 | Linear | ✓ | Sigmoid | [1] |

Table 4: Model Architecture of DCGAN in Section 5.1 and 5.2.

| Net. | Stage | Operation | Stride | Filter Shape | Bias | Norm. | Activation | Output |
|---|---|---|---|---|---|---|---|---|
| - | 0 | Input | - | - | - | - | - | [32] |
| $G$ | 1 | Deconv2D | 1 | [2, 2] | ✓ | ✓ | Sigmoid | [2, 2, $h_G$] |
| $G$ | 2 | Deconv2D | 1 | [3, 3] | ✓ | ✓ | Sigmoid | [4, 4, $h_G$] |
| $G$ | 3 | Deconv2D | 2 | [3, 3] | ✓ | ✓ | Sigmoid | [9, 9, $h_G$] |
| $G$ | 4 | Deconv2D | 1 | [2, 2] | ✓ | ✓ | Sigmoid | [10, 10, $h_G$] |
| $G$ | 5 | Deconv2D | 1 | [3, 3] | ✓ | ✓ | Sigmoid | [12, 12, $h_G$] |
| $G$ | 6 | Deconv2D | 2 | [3, 3] | ✓ | ✓ | Sigmoid | [25, 25, $h_G$] |
| $G$ | 7 | Deconv2D | 1 | [4, 4] | ✓ | ✓ | Sigmoid | [28, 28, $h_G$] |
| $G$ | 8 | Conv2D | 1 | [1, 1] | ✓ | - | Tanh | [28, 28, 1] |
| $D$ | 9 | Conv2D | 1 | [4, 4] | ✓ | ✓ | Sigmoid | [25, 25, $h_D$] |
| $D$ | 10 | Conv2D | 2 | [3, 3] | ✓ | ✓ | Sigmoid | [12, 12, $h_D$] |
| $D$ | 11 | Conv2D | 1 | [3, 3] | ✓ | ✓ | Sigmoid | [10, 10, $h_D$] |
| $D$ | 12 | Conv2D | 1 | [2, 2] | ✓ | ✓ | Sigmoid | [9, 9, $h_D$] |
| $D$ | 13 | Conv2D | 2 | [3, 3] | ✓ | ✓ | Sigmoid | [4, 4, $h_D$] |
| $D$ | 14 | Conv2D | 1 | [3, 3] | ✓ | ✓ | Sigmoid | [2, 2, $h_D$] |
| $D$ | 15 | Conv2D | 1 | [2, 2] | ✓ | ✓ | Sigmoid | [1, 1, $h_D$] |
| $D$ | 16 | Linear | - | - | ✓ | - | Sigmoid | [1] |

Table 5: Hyper parameters in Section 5.2.

| | $K$ | $\eta_G^{[t]}$ | $\eta_D^{[t]}$ | $N$ | $N'$ | $N_{test}$ | $\mathcal{S}_t$ | $\gamma$ | $h_G$ | $h_D$ |
|---|---|---|---|---|---|---|---|---|---|---|
| 2D-Normal | 70 | $10^{-3}$ | $10^{-3}$ | 50k | 10k | 10k | 100 | $10^{-3}$ | 32 | 64 |
| MNIST | 20 | $10^{-3}$ | $10^{-3}$ | 50k | 10k | 10k | 100 | $10^{-3}$ | 32 | 32 |

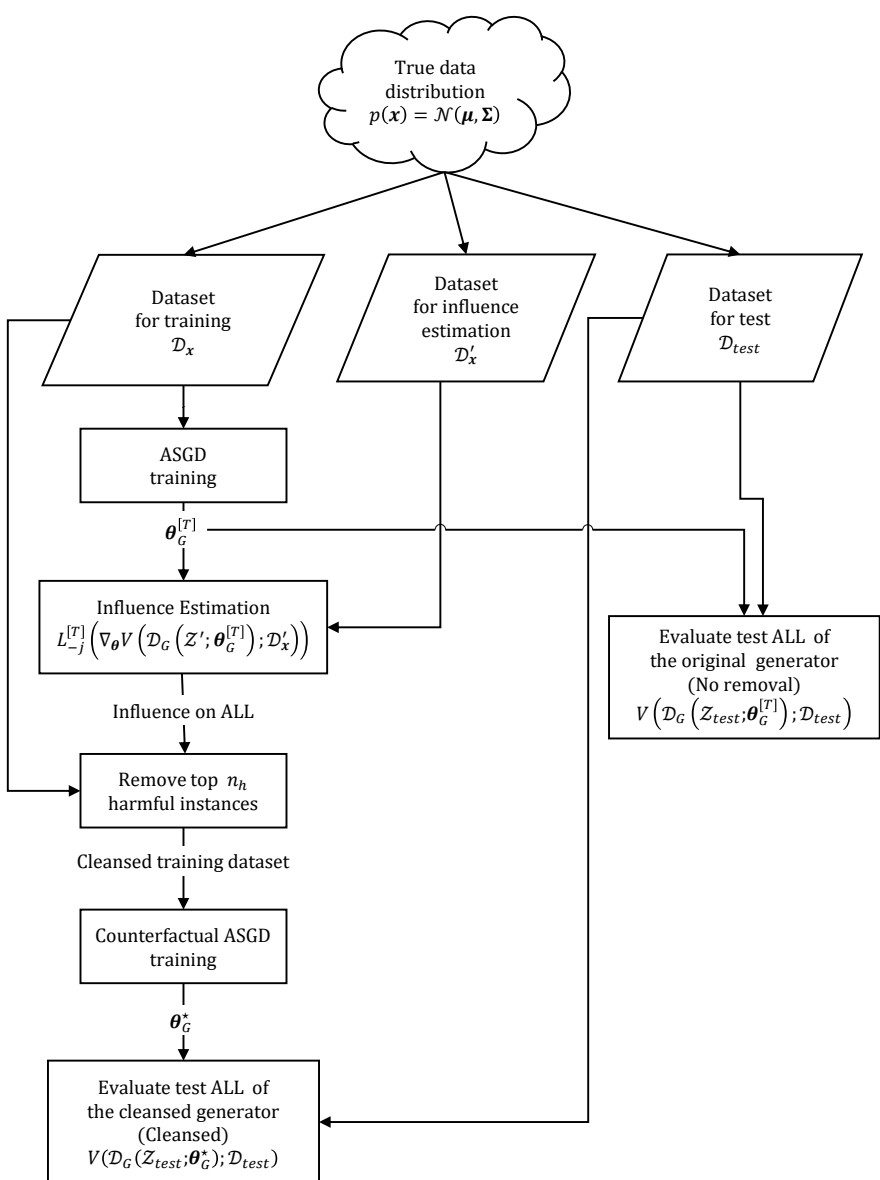

Figure 5: The data setting of data cleansing with the influence on ALL (2D-Normal) in Section 5.2.

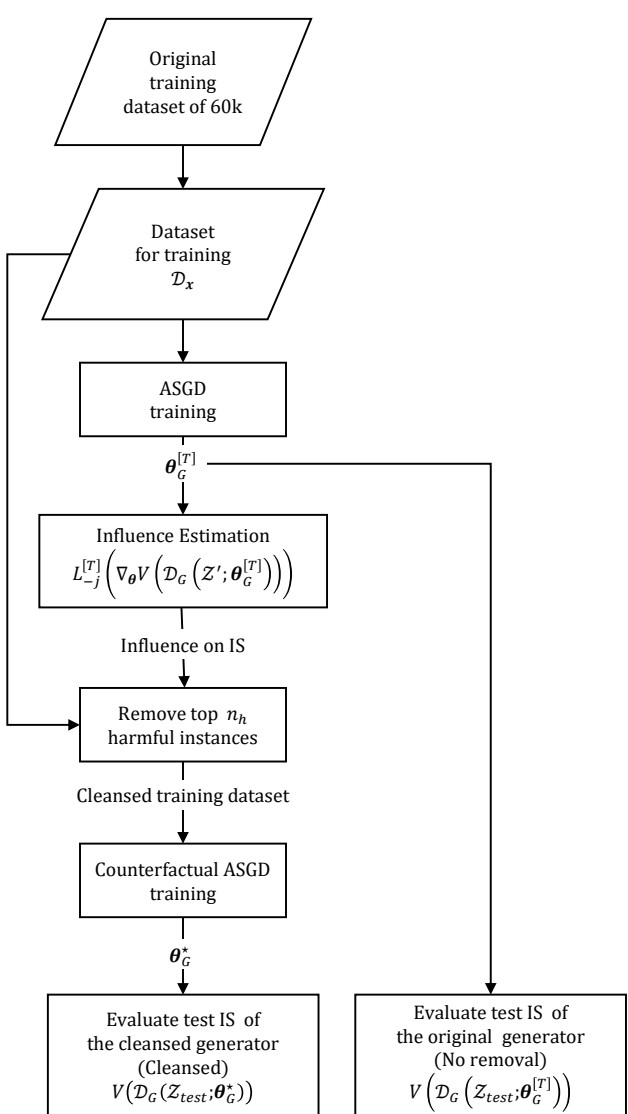

Figure 6: The data setting of data cleansing with the influence on IS (MNIST) in Section 5.2.

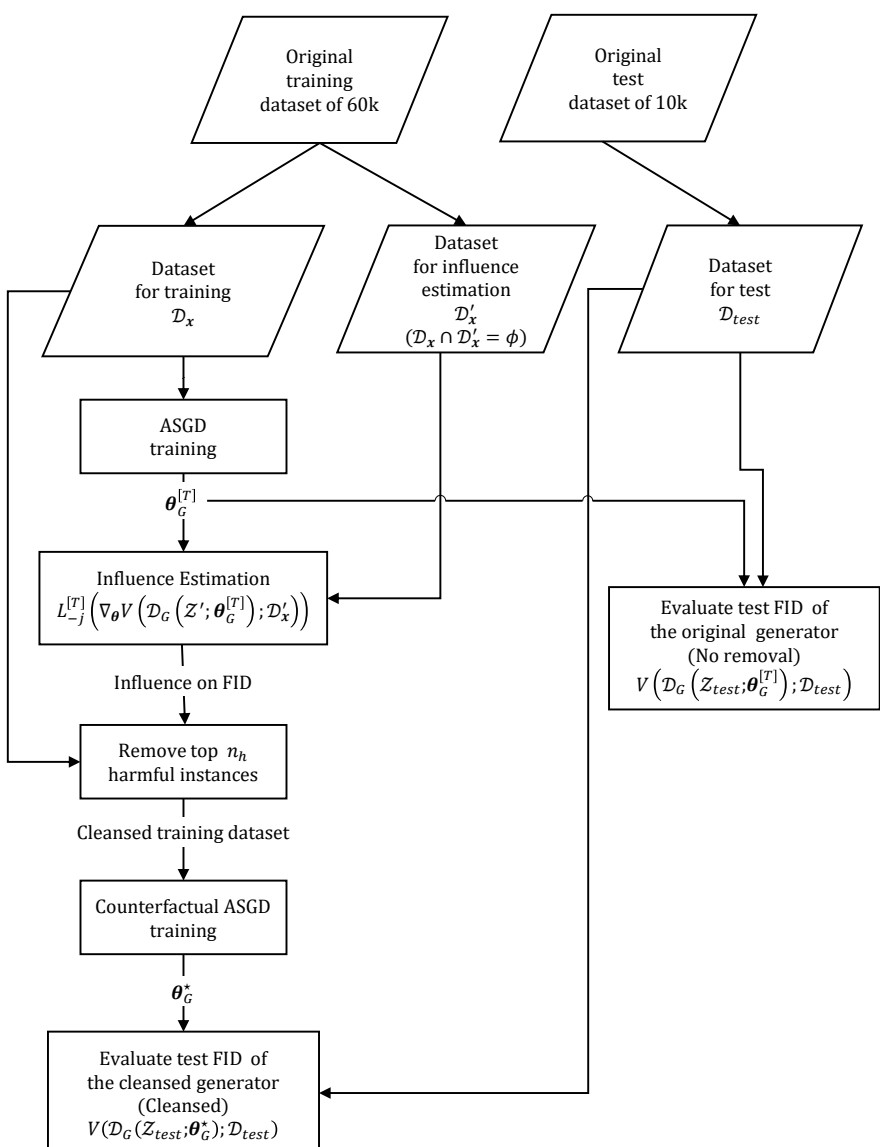

Figure 7: The data setting of data cleansing with the influence on FID (MNIST) in Section 5.2.

Table 6: Improvements of test average log-likelihood $[10^{-2}]$ ($\pm$std) after the data cleansing (2D-Normal). The metric value is highlighted when the improvement is statistically significant with the significant level 0.05

| | $n_h$ | | | | | | | | | |
|---|---|---|---|---|---|---|---|---|---|---|
| | 0.5k | 1.0k | 2.5k | 5.0k | 7.5k | 10.0k | 12.5k | 15.0k | 17.5k | 20.0k |
| Influence on ALL | **+0.09** **(0.06)** | **+0.16** **(0.12)** | **+0.31** **(0.27)** | **+0.44** **(0.50)** | +0.40 (0.73) | +0.22 (0.99) | -0.10 (1.28) | -0.53 (1.60) | -1.07 (1.95) | -1.67 (2.33) |
| Influence on Disc. loss | **+0.02** **(0.03)** | **+0.04** **(0.05)** | **+0.11** **(0.10)** | **+0.19** **(0.19)** | **+0.26** **(0.28)** | **+0.32** **(0.39)** | **+0.35** **(0.51)** | +0.35 (0.64) | +0.30 (0.79) | +0.22 (0.95) |
| Isolation Forest | +0.03 (0.05) | +0.05 (0.11) | +0.09 (0.27) | +0.12 (0.54) | +0.12 (0.79) | +0.09 (1.05) | +0.02 (1.31) | -0.09 (1.58) | -0.25 (1.86) | -0.46 (2.16) |
| Random | +0.01 (0.04) | +0.02 (0.08) | +0.04 (0.19) | +0.07 (0.39) | +0.08 (0.61) | +0.06 (0.83) | +0.02 (1.07) | -0.05 (1.34) | -0.16 (1.61) | -0.31 (1.91) |

Table 7: Improvements of test inception score ($\pm$std) after the data cleansing (MNIST). The metric value is highlighted when the improvement is statistically significant with the significant level 0.05

| | $n_h$ | | | | | | | | | |
|---|---|---|---|---|---|---|---|---|---|---|
| | 0.5k | 1.0k | 2.5k | 5.0k | 10.0k | 15.0k | 20.0k | 25.0k | 35.0k | 45.0k |
| Influence on FID | +0.03 (0.07) | +0.04 (0.09) | +0.04 (0.17) | +0.03 (0.25) | +0.04 (0.24) | **+0.09** **(0.13)** | **+0.10** **(0.12)** | **+0.10** **(0.13)** | +0.04 (0.17) | -0.18 (0.28) |
| Influence on IS | **+0.04** **(0.05)** | +0.04 (0.08) | +0.05 (0.14) | +0.04 (0.23) | +0.08 (0.15) | **+0.11** **(0.13)** | **+0.12** **(0.14)** | **+0.14** **(0.14)** | +0.09 (0.25) | -0.07 (0.24) |
| Influence on Disc. Loss | +0.01 (0.03) | +0.01 (0.05) | **+0.02** **(0.03)** | **+0.04** **(0.04)** | **+0.04** **(0.05)** | **+0.04** **(0.06)** | **+0.04** **(0.06)** | +0.01 (0.06) | +0.00 (0.07) | -0.15 (0.11) |
| Isolation Forest | +0.00 (0.02) | +0.01 (0.02) | +0.01 (0.04) | +0.00 (0.05) | -0.01 (0.06) | -0.05 (0.08) | -0.13 (0.13) | -0.23 (0.18) | -0.67 (0.33) | -1.70 (0.75) |
| Random | +0.01 (0.02) | +0.00 (0.01) | +0.00 (0.02) | -0.01 (0.04) | +0.00 (0.04) | +0.00 (0.05) | -0.01 (0.09) | +0.00 (0.06) | -0.02 (0.07) | +0.00 (0.10) |

Table 8: Improvements of test FID ($\pm$std) after the data cleansing (MNIST). The metric value is highlighted when the improvement is statistically significant with the significant level 0.05

| | $n_h$ | | | | | | | | | |
| --- | --- | --- | --- | --- | --- | --- | --- | --- | --- | --- |
| | 0.5k | 1.0k | 2.5k | 5.0k | 10.0k | 15.0k | 20.0k | 25.0k | 35.0k | 45.0k |
| Influence on FID | **-0.10** **(0.13)** | **-0.13** **(0.18)** | **-0.18** **(0.28)** | -0.19 (0.46) | -0.25 (0.45) | **-0.36** **(0.35)** | **-0.38** **(0.36)** | **-0.38** **(0.37)** | -0.23 (0.46) | +0.23 (0.60) |
| Influence on IS | **-0.07** **(0.10)** | **-0.10** **(0.14)** | **-0.14** **(0.22)** | -0.14 (0.37) | **-0.26** **(0.28)** | **-0.32** **(0.29)** | **-0.34** **(0.30)** | **-0.36** **(0.30)** | -0.22 (0.45) | +0.17 (0.49) |
| Influence on Disc. Loss | -0.03 (0.06) | -0.04 (0.08) | **-0.07** **(0.07)** | **-0.13** **(0.10)** | **-0.18** **(0.12)** | **-0.20** **(0.13)** | **-0.19** **(0.14)** | **-0.15** **(0.14)** | -0.06 (0.12) | +0.34 (0.19) |
| Isolation Forest | +0.01 (0.03) | +0.02 (0.03) | +0.05 (0.06) | +0.10 (0.08) | +0.24 (0.15) | +0.42 (0.22) | +0.73 (0.37) | +1.09 (0.54) | +2.56 (0.85) | +6.99 (3.57) |
| Random | -0.01 (0.04) | -0.01 (0.02) | -0.00 (0.04) | +0.01 (0.06) | +0.00 (0.07) | +0.01 (0.08) | +0.02 (0.16) | -0.01 (0.09) | +0.00 (0.13) | **-0.13** **(0.18)** |

# D  DETAILED DISCUSSION ON EXPERIMENT 2

This section first discusses three aspects of the results in Section 5.2: Section D.1 explains the common characteristics of harmful instances suggested by our approach, Section D.2 discusses qualitative aspects of the data cleansing using generated samples, and Section D.3 discusses how the characteristics of harmful instances and effect of the data cleansing are consistent among the trainings with different random seeds. Finally, we explain the limitation of our method and present the future direction in Section D.4.

## D.1  CHARACTERISTICS OF HARMFUL INSTANCE

In this section, we examine the characteristics of instances that are evaluated to be harmful or *helpful* by our method. We regard a sample is helpful if its influence on a metric is opposite of harmful instances.

Table 9 shows the estimated harmfulness of the training instances of 2D-Normal and the distribution of the generated samples. The proposed approach with influence on ALL evaluated the instances around lower-left and upper-right regions to be harmful (Table 9 (a, i)). These regions correspond to the regions where the generated distribution has higher density than that of the true distribution; The generator before the cleansing (Table 9 (a, ii, No removal)) sampled too frequently from lower-left and upper-right regions compared to the true distribution (Table 9 (a, ii, True)). This characteristics was not observed in the plots of baseline approaches. The approach based on influence on the discriminator loss seems to ignore the difference in the density around the lower-left region (Table 9 (b, i)) and isolation forest did not take the generator's distribution into account (Table 9 (c, i)).

Similar characteristics were seen in harmful MNIST instances suggested by our approach with influence on IS and FID. When the generator over-sampled a specific digit (e.g., the digit 1 in Table 10 (a, iii)), our approach tended to judge the images of the digit to be harmful (e.g., a large number of 1 in Table 10 (b-c, i)). Similarly, our method judged instances of a specific digit as helpful (e.g., the digit 6 in Table 10 (b-c, ii)) when the generator failed to sample the digit (e.g., the absence of 6 in Table 10 (a, iii)). On the contrary, harmful instances suggested on the basis of influence on the discriminator loss did not show the tendency (Table 10 (d, i)). The baseline approach with isolation forest based on the classifier feature-space seems to have judged the images that were difficult to be classified as harmful, rather than the over-sampled digit (Table 10 (e, i)). It regarded that instances are helpful when they belong to a digit that seems to have been easy to be classified (Table 10 (e, ii)).

To summarize, our method tends to judge instances as harmful when they belong to regions from which the generators sample too frequently compared to the true distribution.

## D.2 QUALITATIVE STUDY OF DATA CLEANSING

We then investigate how the data cleansing using the suggested harmful instances visually change generated samples.

As seen from Table 9 (a, ii), the probability density in the upper-right region decreased after the data cleansing (from "No removal" to "Cleansed"). As a result, the generator distribution got closer to the true distribution. Although the baselines indicated the same direction of changes in the distributions (Table 9 (b-c, ii)), these were not as significant as ours.

The same effect was observed in visually more interesting form in the data cleansing for MNIST. The generated samples originating from some latent variables changed from the image of digit 1 to that of other digits after the data cleansing based on the estimated influence on IS and FID (highlighted samples in Table 10 (b-c, iii)). This implies that a certain amount of density that are over-allocated for the digit 1 moved to the regions of other digits. We assume this effect improved the diversity in the generated samples, resulting in better FID and IS. This characteristics was not clearly observed in the baselines (highlighted samples in Table 10 (d-f, iii)).

These observations suggest that our method helps the GAN's training so that the generator re-assigns the densities that were over-allocated to certain regions to other regions.

## D.3 CONSISTENCY OF QUALITATIVE CHARACTERISTICS AMONG DIFFERENT TRAININGS

We show additional visual results to confirm the consistency of the findings on the characteristics of harmful instances and generated samples after data cleansing, which we described in Section D.2 and Section D.3, respectively.

Table 11 shows the harmfulness of the training instances and the distribution of the generated samples obtained using 5 different random seeds in 2D-Normal case. As seen from the table, regardless of which region a generator assigns high density to, our method consistently regards the training samples around the region as harmful. In addition, the distributions of the generated samples get closer to the true distribution by removing these harmful training instances in the data cleansing.

Table 12 visualizes the MNIST examples of harmful instances, helpful instances, and generated images before and after the data cleansing. Different rows correspond to different random seeds. We found the consistency in visual characteristics was moderate in MNIST case. A few results demonstrated the common qualitative characteristics when the improvements in GAN evaluation metrics were large (Table 12 (a) and (d)). In the training with the 4th random seed (d), the suggestion of harmful instances showed some tendency; many instances of digit 7 were regarded as harmful whereas those of digit 4 were not at all (Table 12 (d, i)). The data cleansing based on this suggestion seems to have improved the diversity of the generated samples by reducing the samples of digit 7 and increasing those of digit 4 (highlighted samples in Table 12 (d, iv)). This indicates the consistent characteristics of the data cleansing discussed in the previous section to some extent; it helps the GAN's training so that the generator re-assigns the densities that were over-allocated to certain data regions to other regions.

## D.4 CURRENT LIMITATION AND FUTURE DIRECTION

The limitation of our method is that it does not guarantee the harmful instances suggested on the basis of influence on one GAN evaluation metric are not necessarily harmful from the viewpoint of other metrics. For example, we have demonstrated that removing instances that predicted to have negative influence on FID improved both test FID and IS (Figure 2) and increased visual diversity in generated images (Table 10 and 12). However, it does not seem to have improved visual quality (e.g., sharpness, reality, etc.) of the individual generated-samples. Therefore, it is possible that these instances are harmful only for some particular aspects of generative performance, i.e. the diversity in this case, and they are not harmful for the other aspect, i.e. the visual quality in this case.

We would argue that this limitation is closely tied with the limitation of the current GAN evaluation metrics. For example, FID takes the diversity of generated samples into account, but they only partly take the visual quality into account; e.g., FID based on Inception Net was shown to focus on textures rather than shapes of the objects (Karras et al. (2020)). In this sense, we clarify that we never claim our method can improve the *"true"* generative performance from all the aspects, considering the situation that there is no *"true"* evaluation metric that measures all the aspects of the generative performance.

The advantage of our method is that it does not have to care how the evaluation metrics are defined as long as they are differentiable with respect to the generated samples. Furthermore, our evaluation method makes no assumption about what the harmful characteristics of instances are. This means that it is expected to be easily applied to another evaluation metric if better metric is developed in the future. One of our main contributions in such sense is that we experimentally verified that our method successfully improved the generative performance in terms of a targeted metric, using limited but currently widely accepted metrics.

Our future work includes incorporating such future improvements in the GAN evaluation metric to obtain better insights on the relationship between training instances and generative performance. In addition, we would like to relax the current constraint on the optimizer. Our method is currently applicable only to SGD but we would like to find a way to extend it to other optimizers such as Adam (Kingma & Ba (2015)) to deal with the latest GAN models.

Table 9: (i) harmfulness of 2D-Normal instances suggested by different approaches, (ii) changes in the generator's distribution, and (iii) test ALL after the data cleansing. (ii) includes plots of the true distribution (True) and generator's distributions before (No removal) and after (Cleansed) the data cleansing with $n_h = 5.0$k. The distributions of generated samples, that refer to $\mathcal{D}_G(\mathcal{Z}_{test}; \boldsymbol{\theta}_G^{[T]})$ (No removal) and $\mathcal{D}_G(\mathcal{Z}_{test}; \boldsymbol{\theta}_G^{\star})$ (Cleansed), are estimated with kernel density estimation.

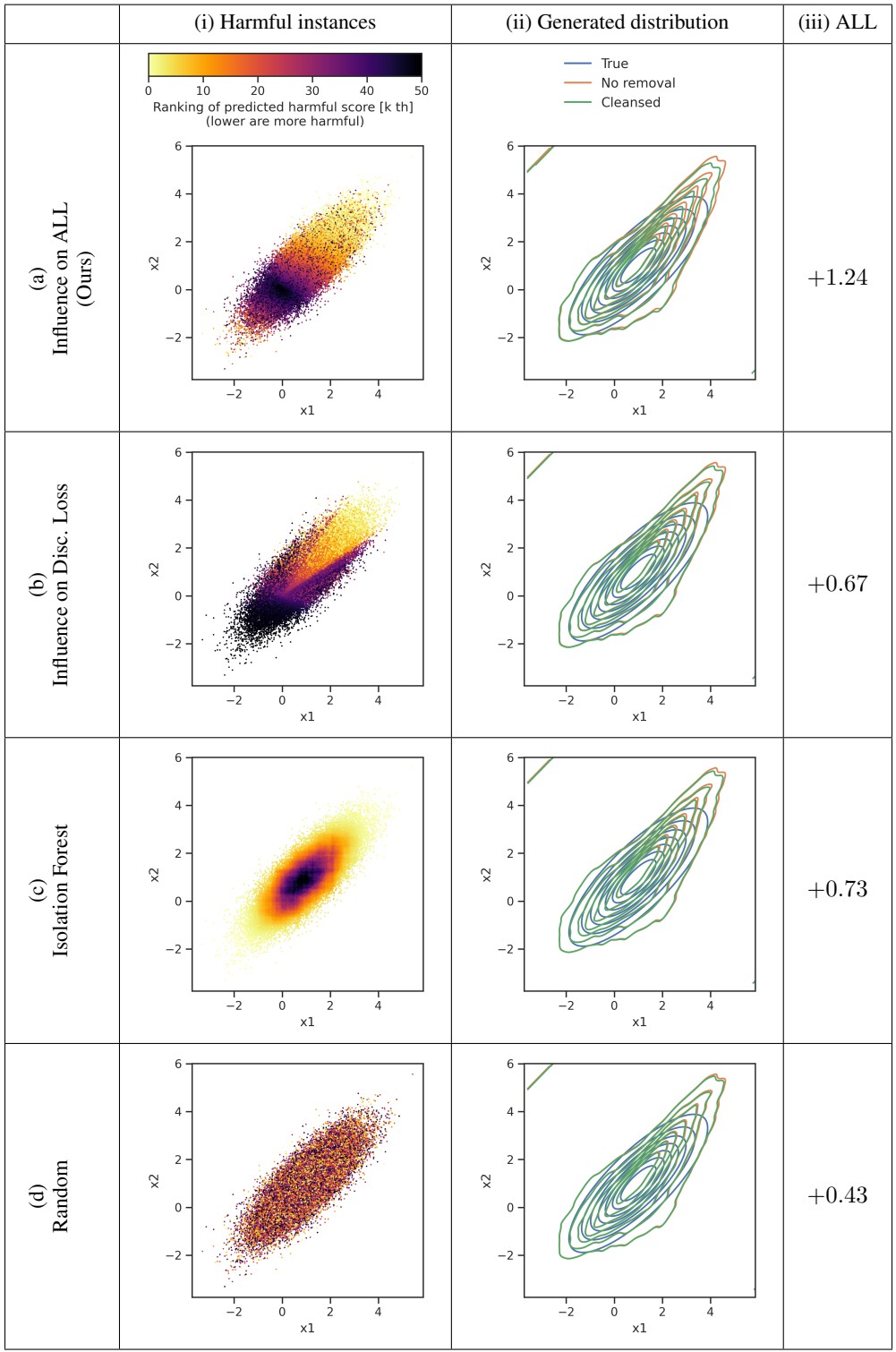

Table 10: (i) top 36 harmful and (ii) helpful MNIST instances predicted by the different approaches, (iii) the test generated samples, and (iv) changes in test FID after the data cleansing with $n_h = 25.0k$. All the generated samples use the same series of test latent variables in $\mathcal{Z}_{test}$.

| | (i) Harmful | (ii) Helpful | (iii) Generated | (iv) FID |
|---|---|---|---|---|
| (a) No removal | n/a | n/a |  | $\pm 0$ |
| (b) Influence on IS (Ours) |  |  |  | $-0.71$ |
| (c) Influence on FID (Ours) |  |  |  | $-0.85$ |
| (d) Influence on D Loss |  |  |  | $-0.21$ |
| (e) Isolation Forest |  |  |  | $+1.80$ |
| (f) Random |  |  |  | $-0.21$ |

Table 11: Comparison among different random seeds used in the training in 2D-Normal case. See Table 9 for how the plots are generated.

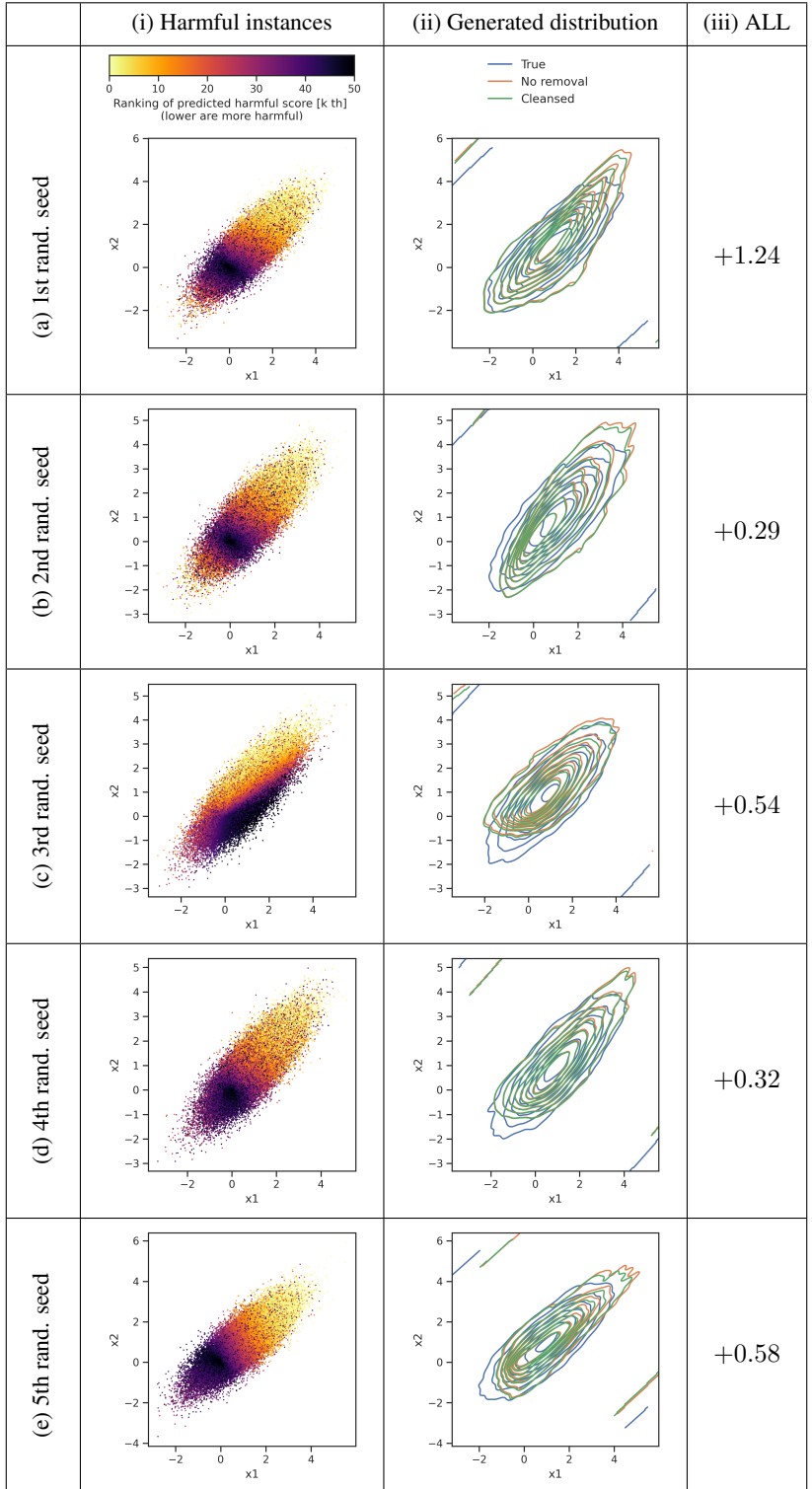

Table 12: Comparison among different random seeds used in the training in MNIST case. The generated samples from the model without cleansing (iii) and cleansed model (iv) in the same row use the same series of test latent variables. See Table 10 for the detail of how the images are obtained.

| | (i)
Harmful | (ii)
Helpful | (iii)
Generated
(No removal) | (iv)
Generated
(Cleansed) | (v)
FID |
|---|---|---|---|---|---|
| (a) 1st rand. seed | | | | | −0.85 |
| (b) 2nd rand. seed | | | | | −0.45 |
| (c) 3rd rand. seed | | | | | +0.09 |
| (d) 4th rand. seed | | | | | −0.71 |
| (e) 5th rand. seed | | | | | −0.12 |

