# OpenReview forum: "Influence Estimation for Generative Adversarial Networks"
_ICLR.cc/2021/Conference — ICLR 2021 Spotlight_

### Official Review · AnonReviewer2 · 2020-10-23
**Provides a detailed theoretical and experimental analysis of the proposed approach on influence estimation for GANs, but there are concerns regarding its practical applicability.**

**Rating:** 7
**Confidence:** 3

**Review:**

Summary:

The paper presents an influence estimation method for GANs. It discusses why previous approaches on influence estimation cannot be easily extended to GANs. It proposes to use Jacobian of the gradient of discriminator’s loss with respect to the generator’s parameters to learn how absence of an instance in the discriminator’s training affects the generator’s parameters. The authors evaluate whether an instance is harmful based on its influence on GAN evaluation metrics. They show that removing these harmful instances improves performance of GANs on MNIST with respect to three metrics: Inception Score, FID and Average Log Likelihood (ALL).

#################################################################

Strengths:

1. The paper is well-written, motivates the problem well and provides a detailed analysis of the proposed algorithm. It distinguishes its approach with other works on influence estimation such as [Hara et al.] and [Koh & Liang].
2. Several experiments are performed to validate the authors’ claims. They verify that estimated and true influence on GAN evaluation metrics have statistically significant correlation, and show how removing harmful instances can improve evaluation scores.

#################################################################

Weaknesses:

1. For the improvements on evaluation metrics to be significant, the authors need to remove a large number of samples from the dataset. For example, considering the Inception Scores in Table 7, they need to remove 15k instances out of 60k MNIST samples to get statistically significant improvements. Even after removing those instances, improvements are modest (e.g. +0.11/5.8 = 1.9%). It seems that this approach eliminates modes of the distribution that the GAN is not able to cover well. While this can improve the evaluation metrics, this improvement is artificial and does not correspond to a better GAN model. In general, removing underrepresented and rare samples can increase evaluation scores, but a strong GAN model should be able to generate those samples as well. The authors can discuss the nature of harmful samples and whether removing them actually leads to better generation.
2. There are no qualitative results to demonstrate which instances are harmful in training GANs. [Koh & Liang] provide qualitative results about which samples help (Figure 4) and harm (Figure 1) the model’s performance. Considering the large number of samples that need to be removed for improvements in evaluation scores, the authors can examine those samples, visualize them and try to find common characteristics among them.
3. The authors use Isolation Forests [Liu et al. 2008] as the baseline for anomaly detection. It would be better to consider more recent approaches for anomaly detection such as those outlined in [B].

#################################################################

Reason for Rating:

The authors motivate the problem well and provide a detailed theoretical and experimental analysis. However, there are concerns regarding practical applicability of the proposed approach and whether it can indeed help with performance of GANs or it artificially boosts the evaluation scores. There are no qualitative evaluation on which samples harm the model’s performance.

#################################################################

Additional Comments:

There are other metrics for evaluating performance of GAN models such as Precision and Recall [A]. While I think the current metrics considered in the paper are adequate, the authors can also mention other metrics.

##################################################################

References:

[A] Are GANs Created Equal? A Large-Scale Study; Lucic et al.; NeurIPS 2018

[B] Deep Learning for Anomaly Detection: A Survey; Chalapathy et al.; arXiv 2019

##################################################################

After author response: I thank the authors for the additional experiments. It partially addresses my concerns. However, the qualitative results do not always show improvements in image quality, and most of the low-quality samples are still generated after cleansing. It’s also hard to clearly notice differences before and after cleansing in Tables 9 and 11 (2D-Normal). It would be better to use a multi-modal Gaussian with some modes being more likely. Overall, I am still concerned about practical applicability of the proposed approach. I keep my score of 7.

---

> ### Author Response · Authors · 2020-11-23
> **Response to AnonReviewer2 (1/2)**
>
> We thank the reviewer for the constructive comment.
> We think the points raised by the reviewer are quite important and helpful to improve our paper.
>
> We added the discussion on the raised points in Appendix D.
> We will later pick up key parts of the discussion as well as some of the visual examples, and move them to the main paper.
> For now, we prioritize the quick response, and thus place all the additional discussion in Appendix D.
>
> **- Response to weakness 1:**
>
> Thank you for pointing out this important aspect.
> As suggested, we discussed the nature of harmful samples (Section D.1 in the appendix), how the data cleansing based on the harmful samples changes the generated samples (Section D.2), and the limitation of our study (Section D.4).
>
> As far as we examined through the examples of harmful instances, our method did not eliminate modes of the distribution, rather it seems to have increased diversity in the generated samples by removing instances in the specific region from which the generator was over-sampling.
> We suppose this led to *''better''* GAN in terms of the diversity, and resulted in enhancing another metric that was not used for influence estimation; data cleansing based on influence on FID improved test IS (and vice versa).
> However, we do not insist that improving a specific metric (e.g., FID) is always meaningful for all the aspects of generative performance because the metrics we used in the experiment are not perfect.
> For example, FID based on Inception Net was shown to focus on textures rather than shapes of the objects [C].
> As we discussed in Section D.4, the improvement brought by our method is closely tied with the GAN evaluation metrics of your choice.
> One of our main contributions in such sense is that we experimentally verified that our method successfully improved the generative performance in terms of a few metrics, that are limited but currently widely accepted metrics.
>
> We note that the advantage of our method is that it does not have to care how the evaluation metrics are defined as long as they are differentiable with respect to the generated samples. Furthermore, our evaluation method makes no assumption about what the harmful characteristics of instances are. This means that the proposed method is expected to be easily applied to another evaluation metric if a better metric is developed in the future.
>
> **- Response to weakness 2**
>
> As suggested, we added qualitative results in Appendix D, and analyzed the common characteristics among them.
> To summarize, the harmful instances identified by our method tend to belong to the regions from which a generator samples too frequently compared to the true distribution.
>
> As seen in Table 9 (i), the distribution of the training instances of 2D-Normal well represents the true distribution (*True* in Table 9 (ii)) and does not seem biased.
> However, the distribution of the data generated by the trained GAN seems to be biased for some reasons; the generator samples too frequently from the tail region of the true distribution (*No removal* in Table 9 (ii)).
> Our method regards the training instances that are located around this region as harmful (Table 9 (a, i)) probably because these instances influence the GAN model in such a way that it generates too many samples in this region.
> The generator's distribution after the data cleansing demonstrates that removing those instances actually suppressed samples from this region, resulted in making the generator's distribution closer to the true distribution (*No removal* to *Cleansed* in Table 9 (a, ii)).
>
> The same tendency was observed in MNIST case.
> When a generator generates samples of a specific digit too much, our method tends to regard the training instances of the digit as harmful.
> By removing these harmful training instances, the generator tends to generate the samples of the digit less frequently.
>
> Overall, we find that our model tends to regard the training instances as harmful if they belong to the regions around which a generator samples too much.
> The data cleansing based on the identified harmful instances tends to lead to alleviating the undesirable bias that a trained generator has.
>
> We note that the ''*undesirableness*'' may not always align with our intuition (we think it did in our experiment, though).
> We think our method implicitly measures the ''*undesirableness*'' with respect to GAN evaluation metrics used for estimating the harmfulness.
> In such sense, the tendency of harmful training instances identified by our method would change if used GAN evaluation metrics change.
> As mentioned above, we believe our method will benefit from the future development of GAN evaluation metrics.

---

> > ### Author Response · Authors · 2020-11-23
> > **Response to AnonReviewer2 (2/2)**
> >
> > **- Response to weakness 3:**
> >
> > Thank you for your suggestion.
> > Considering the recent advancement in anomaly detection techniques, we agree that the raised point is a part that can be improved.
> >
> > Regarding the selection of baseline methods, here, we would like to add a point according to the observation in the additional qualitative study in Appendix D.
> > The results of our method indicate that the harmfulness of the training instances is strongly connected to the characteristics of the trained model: e.g., from which region the generator over-samples.
> > However, no matter which anomaly detection method we choose, it would not take the generative tendency of the model into account.
> > We would argue that the effectiveness of anomaly detection for the purpose of data cleansing is limited as they are not designed for evaluating the harmfulness of instances to the training.
> > In this view, we hope another method will be proposed in the future that is designed for data cleansing and that enables us to more fairly evaluate the effectiveness of our method.
> >
> > **- Response to additional comment:**
> >
> > We appreciate the suggestion about another evaluation metric such as Precision and Recall, which can separately evaluate 2 aspects of generative performance: quality and coverage of samples.
> > We find that the suggested metrics are helpful to further investigate the additional observation in Appendix D.
> > More precisely, we hope they will clarify whether the improvement in visual diversity of MNIST actually improved the coverage of the generated samples, or not.
> > Although we lack time for additional experiments during the discussion period, we will include the discussion of this aspect in the final version.
> >
> > [C] Tero Karras, Samuli Laine, Miika Aittala, Janne Hellsten, Jaakko Lehtinen, and Timo Aila. Analyzing and improving the image quality of stylegan. In Proceedings of the IEEE/CVF Conference on Computer Vision and Pattern Recognition, pp. 8110–8119, 2020.

---

### Official Review · AnonReviewer1 · 2020-10-26
**Good, but missing a key piece**

**Rating:** 7
**Confidence:** 3

**Review:**

Summary:
This paper introduces influence estimation for GANs. The influence estimates approximate how helpful or harmful each training sample is with respect to some evaluation metric or loss function, such as Inception Score or FID. Removing harmful instances is shown to improve GAN performance.

Strengths:
-Paper is well written.
-Influence estimations adds an interpretability tool to GAN training. This is very welcome, because the training dynamics of GANs are still not very well understand.
-Proposed technique does a good job of estimating the true influence.
-Predicted influences can be used to remove harmful data points, which is shown to improve GAN performance.

Weaknesses:
-No visual examples of the data points that were highly influential.
-No analysis into what kinds of characteristics might make a data point highly influential, either in the helpful or harmful sense.
-None of the evaluation metrics used differentiate between fidelity and diversity. This is particularly important because removing instances from the training set inevitably reduces diversity, but this is not reflected in the current metrics. Including metrics such as Precision and Recall [1] or Density and Coverage [2] would allow us to see the trade-off between fidelity and diversity.
-Computationally expensive, as model parameters need to be saved at every iteration (not a major issue for this paper, but worth noting)
-Proposed method is only applied to very simple datasets and GAN models (likely due to the aforementioned compute issue)

Recommendation and Justification:
I really like the direction that this paper is headed in, but I think there is one thing holding it back. My greatest disappointment with this paper was that it does not include any visual examples of highly influential data points, nor any analysis or discussion about what kinds of characteristics helpful or harmful data points might have. These insights are half of the reason why interpretability methods such as influence estimation are useful, and without them it feels like I have read a story without a satisfying conclusion. In its current state I would rate this paper as being marginally below the acceptable threshold, but if the above concerns were adequately addressed I would very gladly increase my score.

Clarifying Questions:
-In the paper it is stated that the generator and discriminator are simultaneously updated in one step for simplicity. However, the majority of GANs trained in practice use separate update steps for the generator and discriminator. Does this method also work for separate update steps?
-How consistent is the selection of harmful instances? For example, does the set of the most harmful instances change throughout training (at the beginning vs towards the end)? What about across different runs with different random seeds?
-Recently it has been shown that removing data points that lie in low density regions of the data manifold can result in improved GAN performance [3]. I would be curious to know if you think there is any correlation between the harmful instances identified by influence estimation and the density of the data manifold?

[1] Kynkäänniemi, Tuomas, et al. "Improved precision and recall metric for assessing generative models." Advances in Neural Information Processing Systems. 2019.
[2] Naeem, Muhammad Ferjad, et al. "Reliable Fidelity and Diversity Metrics for Generative Models." arXiv preprint arXiv:2002.09797 (2020).
[3] DeVries, Terrance, Michal Drozdzal, and Graham W. Taylor. "Instance Selection for GANs." arXiv preprint arXiv:2007.15255 (2020).

------------------------------------------------------

Edit after author response:
The authors have sufficiently addressed my concerns with the additions that have been added in Appendix D. As such, I will increase my score from a 5 to a 7.

---

> ### Author Response · Authors · 2020-11-20
> **Initial response to AnonReviewer1**
>
> We thank the reviewer for the constructive comment.
> We think the points raised by the reviewer are quite important and helpful to improve our paper.
>
> We added the discussion on the raised points in Appendix D.
> We will later pick up key parts of the discussion as well as some of the visual examples, and move them to the main paper.
> For now, we prioritize the quick response, and thus place all the additional discussion in Appendix D.
>
> In this reply, we first discuss the following point, which we think is the most important one.
> We will also discuss the other raised points including the clarifying questions as soon as possible in a separate reply.
>
> *- Recommendation and justification: Adding visual examples and analysis on the characteristics of harmful instances*
>
> In this reply, we just pick up the key insights on the characteristics of harmful instances identified by our method and make a point of the future direction of our work including the aspect of the interpretability.
> Please refer to the Appendix D for more details and visual results.
>
> As suggested, we added visual examples of data that our method regarded as highly harmful in Table 9 - 12 in Appendix D.
> In addition, we added examples of ''*helpful*'' instances, too, for the reference.
> Please note that we regard a sample is ''*helpful*'' if its influence on a metric is opposite of harmful instances.
>
> The harmful instances identified by our method tend to belong to the regions from which a generator samples too frequently compared to the true distribution.
> The true distribution refers to the distribution that produces the training data $D_x$, validation data $D_x'$, and test data in the paper.
> More concretely, it refers to the distribution that has the true parameters $(\boldsymbol{\mu}, \boldsymbol{\Sigma})$ in 2D-Normal case,
> and the underlying true distribution that is unknown in MNIST case.
>
> As seen in Table 9 (i), the distribution of the training instances of 2D-Normal well represents the true distribution (*True* in Table 9 (ii)) and does not seem biased.
> However, the distribution of the data generated by the trained GAN seems to be biased for some reasons; the generator samples too frequently from the tail region of the true distribution.
> Our method regards the training instances that are located around this region as harmful probably because these instances influence the GAN model in such a way that it generates too many samples in this region.
> The generator's distribution after the data cleansing demonstrates that removing those instances actually suppressed samples from this region, resulted in making the generator's distribution closer to the true distribution (Table 9 (a, ii)).
>
> The same tendency was observed in MNIST case. When a generator generates samples of a specific digit too much, our method tends to regard the training instances of the digit as harmful.
> By removing these harmful training instances, the generator tends to generate the samples of the digit less frequently.
>
> Overall, we find that our model tends to regard the training instances as harmful if they belong to the regions around which a generator samples too much.
> The data cleansing based on the identified harmful instances tends to lead to alleviating the undesirable bias that a trained generator has.
>
> We note that the ''*undesirableness*'' may not always align with our intuition (we think it did in our experiment, though).
> We think our method implicitly measures the ''*undesirableness*'' with respect to GAN evaluation metrics used for estimating the harmfulness.
> In such sense, the tendency of harmful training instances identified by our method would change if used GAN evaluation metrics change.
> The advantage of our method is that it does not have to care how the evaluation metrics are defined as long as they are differentiable.
> Therefore, we believe our method will benefit from the future development of GAN evaluation metrics.
>
> We do agree with your point about interpretability and hope our work will be the initial step of influence-based explanation for GANs that will be further explored in the community.

---

> > ### Author Response · Authors · 2020-11-23
> > **Additional response to AnonReviewer1 (1/2)**
> >
> > We would like to respond to the other raised points in *Weaknesses* and *Clarifying Questions*
> >
> > **- Weakness 1,2: No visual examples and analysis of the harmful instances**
> >
> > We hope the previous comment has already answered these points.
> > We will pick up key examples and analysis in Appendix D and include them in the main paper of the final version to address these weaknesses.
> >
> > **- Weakness 3: None of the evaluation metrics used differentiate between fidelity and diversity**
> >
> > We appreciate the suggestion about other evaluation metrics such as Precision and Recall, which can separately evaluate 2 aspects of generative performance: fidelity and coverage of samples.
> > We find that the suggested metrics are helpful to further investigate the additional observation in Appendix D. More precisely, we hope they will clarify whether the improvement in visual diversity of MNIST actually improved the coverage of the generated samples, or not. Although we lack time for additional experiments during the discussion period, we will include the discussion of this aspect in the final version.
> >
> > **- Weakness 4: Computationally expensive, as model parameters need to be saved at every iteration**
> > **- Weakness 5: Proposed method is only applied to very simple datasets and GAN models (likely due to the aforementioned compute issue)**
> >
> > We would like to answer these two questions together since we think they are closely related to each other.
> > We admit the large storage cost for the parameters could be a bottleneck for practical use, especially when bringing our method to more complex settings.
> >
> > However, we make a point that both the challenges of complex settings and large storage costs are potentially addressed by relaxing the constraint on optimizers.
> > In recent complex settings, GAN models are normally trained with momentum-based optimizers such as Adam.
> > Currently, we constrain the optimizer to be SGD, which makes it difficult to apply our method to those settings.
> > Interestingly, the other challenge of large storage cost for the parameters of steps can also be avoided by using the momentum-based optimizers; it is shown that we do not have to store the parameters if we use Momentum-SGD [A][B].
> > Since our work is extended from the work of [A], we believe there is a good potential that Momentum-SGD can be also leveraged in the influence estimation for adversarial training.
> > Thus, relaxing the constraint on the optimizer would be definitely one of our important future work.
> >
> > **- Question 1: Work with the separated training?**
> >
> > Yes, it works.
> > Our derivations of the estimator still hold even when either of the learning rates takes 0.
> > By taking learning rates such that they alternatively take 0 at each step, we can have the estimator for the separate training.
> > We have also experimentally confirmed that it works as well as the estimator for simultaneous training.
> > We consider this point to be important in practical use, so we will include the above explanation in the final version.
> >
> > **- Question 2: Harmful instances are consistent?**
> >
> > Regarding a raised aspect *'What about across different runs with different random seeds?''*, we observed the consistent characteristics among the different random seeds; they belong to the region to which the generator allocates too high density as discussed in Appendix D.3.
> > We note that the training dataset differs in each random seed in our setting, so we cannot tell how much the same individual instances are suggested among the different experiments.
> >
> > To answer the other aspect *'does the set of the most harmful instances change throughout training?'*, we find that additional implementations are required and they would not be in time for the end of the discussion period.
> > However, we are willing to include the discussion of this aspect in the final version of the paper.

---

> > > ### Author Response · Authors · 2020-11-23
> > > **Additional response to AnonReviewer1 (2/2)**
> > >
> > > **- Question 3: Related to density in the manifold?**
> > >
> > > We find that the suggested point is interesting for understanding what we have done from a different context.
> > >
> > > As far as we observed in the experiment described in Appendix D, our method does not seem to have regarded the data that lie in low density regions as harmful merely because they are in low density regions, rather, it seems to have regarded the data as harmful if they belong to the region where the generator samples too frequently compared to the true distribution.
> > > In other words, it seems to have placed more importance on the difference of densities between the generated distribution and the true distribution rather than the vanilla density of the data distribution.
> > >
> > > In the work [3] cited by the reviewer, they say
> > > >we remove low density regions from the data manifold prior to model optimization and show that this direct dataset intervention improves overall image sample quality in exchange for some reduction in diversity.
> > >
> > > Our method attempts to make the generated distribution get closer to the true distribution by using a targeted metric as an indicator of the closeness.
> > > As such, our method strongly reflects on which aspect the targeted metric places emphasis on.
> > > Therefore, there may be some cases where our method behaves similarly to the work of [3] depending on the choice of the targeted metric.
> > > More concretely, our method would regard the instances of low density regions as harmful if the targeted metric places more importance on the image sample quality.
> > > In this case, the generative performance measured by the targeted metric would be improved by removing the data of low density regions (probably at the sacrifice of diversity as mentioned in [3]).
> > >
> > > We appreciate your suggestion, which is very helpful to contextualize our work in a broader view.
> > > We will discuss the relation to the suggested paper in the final version.
> > >
> > > [A] Satoshi Hara, Atsushi Nitanda, and Takanori Maehara. Data Cleansing for Models Trained with SGD. In Advances in Neural Information Processing Systems 32, 2019.
> > >
> > > [B] Maclaurin, Dougal, David Duvenaud, and Ryan Adams. "Gradient-based hyperparameter optimization through reversible learning." International Conference on Machine Learning. 2015.

---

> > > > ### Comment · AnonReviewer1 · 2020-11-24
> > > > **Review Update**
> > > >
> > > > Thanks to the authors for adding the additional results and analysis, this is exactly what I was hoping to see! The paper feels much more complete now. I have increased my score accordingly.

---

### Official Review · AnonReviewer3 · 2020-10-28
**Paper is interesting to read and think about, but there are no meaningful results**

**Rating:** 6
**Confidence:** 3

**Review:**

The paper proposes a technique to identify "harmful" data samples in the training dataset of a GAN. The research question is if it would be possible to find training samples that can be removed to improve the GAN training.

I found the paper generally well written and interesting to read. I have some experience with GANs, but with so many papers being published in general I am not very confident in my evaluation that something like this hasn't done before. The authors argue that this idea has been applied in other settings and that they are the first to bring this idea to GANs and adapt the overall technical idea to this new context. I agree with the authors that this is technically interesting and a worthwhile topic to explore. It is also helpful to have some papers on this topic and someone has to write the first paper. I am supportive of having some research in this direction and I believe other similar work can follow that builds on this initial idea.

On the downside, the paper fails to establish that the proposed idea can actually contribute to creating a GAN that samples images of better visual quality. I would argue that this should be the main goal. The GAN research is somewhat split. This paper  uses small datasets and builds on a simple architecture (DC GAN) that is very far from a state-of-the-art GAN (e.g. StyleGAN2).

The first question is how can we know if the presented work really transfers to a GAN that is currently relevant. The paper does not offer any insights here and this is a part that could be improved. It would be preferable if such tests would already be performed by the authors so that others can know if it is worth following this direction of research.

Another question is why no images are shown in the paper. All results are using GAN evaluation metrics: FID score, Inception, and ALL. From these metrics, I doubt that ALL is as meaningful as the authors make it out to be and I do not think the Inception score is that great either. FID has been quite useful to compare GANs.
However, I am skeptical that improving the FID score directly is a meaningful endeavor. It's easily possible that an attempt to directly influence a metric such as FID score doesn't lead to better results for GANs. The metric itself has many problems and it is easy to exploit a weakness in the metric when directly optimizing for it. Similarly, an important GAN component, truncation, significantly improves quality while significantly worsening the FID. This is also an indication that the FID score is not a great metric that can be directly optimized for.
Targeting the FID directly also has other technical problems. Removing samples from the training set changes the target mean and covariance statistics of the inception v3 network. The idea of the FID score is to compute a 2048 dimensional mean vector and covariance matrix summarizing a set of samples. Typically, the statistics of the training set and the statistics of generated GAN samples are compared.
In this setup, removing training images changes the problem statement. Training a GAN with removed images can no longer be compared to training a GAN with all the images. What do you do here. Do you use all the original training dataset samples to compute the FID? That doesn't make sense in my view. Then the FID is computed with respect to a different dataset.
Or do you only use the new training dataset to compute the FID? Maybe the FID is better, but you also might have just simplified the problem.
How can we judge the visual results here? I would guess the results are either not distinguishable and all look mediocre (because the baseline network is outdated) or maybe the new results may even look a bit worse. Do we really want to have GAN papers without any images?

---

> ### Author Response · Authors · 2020-11-23
> **Response to AnonReviewer3 (1/2)**
>
> We thank the reviewer for the constructive comment.
> We think the points raised by the reviewer are quite important and helpful to improve our paper.
>
> We added more detailed analysis and discussion on the experimental result especially from qualitative perspective as suggested by the reviewer.
> Please find them in Section D in the appendix.
> We will later pick up key parts of the discussion as well as some of the visual examples, and move them to the main paper.
> For now, we prioritize the quick response, and thus place all the additional discussion in Appendix D.
>
> > On the downside, the paper fails to establish that the proposed idea can actually contribute to creating a GAN that samples images of better visual quality. I would argue that this should be the main goal.
>
> We discussed how the visual characteristics of the generated samples changed through the data cleansing in Section D.2 in the appendix.
> We also discussed the current limitation of our work in Section D.4 in the appendix.
>
> To summarize, we think our method succeeded in improving the *''quality''* of the generated samples in terms of their diversity, but we admit that it did not clearly improve the visual quality of the individual samples (e.g., sharpness or reality of images).
> However, we do not think this means our method is useless.
> As we discussed in Section D.4, the improvement brought by our method is closely tied with the GAN evaluation metrics of your choice.
> Our method is designed to improve the performance measured in a targeted metric, and the metrics we used in the experiment, which are widely used in many papers as well, do not perfectly measure the visual quality.
> For example, FID based on Inception Net was shown to focus on textures rather than shapes of the objects [A]).
> We think the improvement in diversity as well as the unclarity in the improvement in visual quality are attributed to the characteristics of the targeted metric, namely ALL, IS, or FID in our case.
>
> One of the most important advantages of our method is that it does not have to care how the evaluation metrics are defined as long as they are differentiable with respect to the generated samples.
> Furthermore, our evaluation method makes no assumption about what the harmful characteristics of instances are.
> This means that it is expected to be easily applied to another evaluation metric if a better metric is developed in the future.
> One of our main contributions in such a sense is that we experimentally verified that our method successfully improved the generative performance in terms of a few metrics that are limited but currently widely accepted metrics.
>
> > How can we know if the presented work really transfers to a GAN that is currently relevant?
>
> Technically, our work is applicable to the latest architectures that have various components such as conditional inputs, AdaIN, and skip connections, although we have not tried yet.
> This is because our method does not rely on any specific features of a specific architecture.
> The limitation of our work rather lies on the optimizer to be used as discussed in Section D.4 in the appendix.
> We assume the models are trained with SGD, and therefore our method cannot be directly applied to recent GAN models that are commonly trained with other optimizers such as Adam.
> We note that the prior work on influence estimation in supervised learning setting also has this limitation [B].
> Relaxing the constraint on the optimizer would be definitely one of our important future work.
>
> > No images are shown in the paper.
>
> We prioritized the quantitative results in the original manuscript, but we are happy to share the visual results.
> Please find them in Appendix D.
> Please also refer to the aforementioned summary of the characteristics that we found in the visual results.
>
> > Improving FID directly does not lead to better GAN.
>
> We agree with your statement that *''``The metric (FID) itself has many problems''* and the suggested point that improving FID might not lead to better visual quality.
> We think we have already answered to this point earlier in this reply *"``On the downside, ...''*.
> Please also refer to the discussion in Appendix D.4.
>
> Since FID has a weakness as a measure of generative performance as pointed out, targeting FID in the training procedure to improve a GAN model may not be always meaningful especially when the improvement is measured by FID itself.
> However, it is widely accepted in prior works to target FID in the training procedure to improve a GAN model, as long as the improvement is measured using another metric that is not targeted.
> For example, [C] and [D] optimized neural architectures of GANs using IS as the rewards and evaluated the model with FID.
> In our case, we found the data cleansing using influence on FID resulted in improving IS as well (and vice versa).

---

> > ### Author Response · Authors · 2020-11-23
> > **Response to AnonReviewer3 (2/2)**
> >
> > > Removing training images changes the problem statement. ... What do you do here?
> >
> > Thank you for the clarifying question.
> > We evaluated FID using *''test''* datasets which are used for neither training (denoted as $D_x$ in the paper) nor influence estimation (denoted as $D_x'$ in the paper). We added the diagram to clarify the evaluation setting in Figure 3 \~ 5 in Appendix C.3.
> >
> > In our view, the important thing when computing FID is to use a dataset that is expected to represent the statistics of the true distribution.
> > On the one hand, many papers use the training data set to compute FID because the training data set is expected to be sampled from the true distribution, as the reviewer pointed out.
> > On the other hand, there are some cases where a separate dataset (called test set), which is also expected to be sampled from the identical true distribution.
> > For example, Kaneko et al. [E] trained models using training datasets that has artificial noise and computed FID using a separate dataset (test set) without any noises, which is expected to represent the true distribution.
> > Although the statistics of the training set (with noise) is different from that of test set (without noise), we believe their evaluation is valid because what they want to (and should) do is to compare the statistics of generated images with that of images sampled from the true distribution.
> > For the same reason, we used a separate test set (, which we newly denote by $D_{test}$ in Figure 5) to compute FID in our experiment.
> > Therefore we believe our evaluation settings were adequate as it was in [E].
> >
> > [A] Tero Karras, Samuli Laine, Miika Aittala, Janne Hellsten, Jaakko Lehtinen, and Timo Aila. Analyzing and improving the image quality of stylegan. In Proceedings of the IEEE/CVF Conference on Computer Vision and Pattern Recognition, pp. 8110–8119, 2020.
> >
> > [B] X. Gong, S. Chang, Y. Jiang, and Z. Wang, “Autogan:  Neural architecture search for generative adversarial networks,” in Proceedings of the IEEE International Conference on Computer Vision, pp. 3224–3234, 2019.
> >
> > [C] H. Wang and J. Huan, “Agan:  Towards automated design of generative adversarial networks,”arXiv preprint arXiv:1906.11080, 2019.
> >
> > [D] Satoshi Hara, Atsushi Nitanda, and Takanori Maehara. Data Cleansing for Models Trained with SGD. In Advances in Neural Information Processing Systems 32, 2019.
> >
> > [E] Takuhiro Kaneko and Tatsuya Harada. Noise robust generative adversarial networks. In Proceedings of the IEEE/CVF Conference on Computer Vision and Pattern Recognition, pp. 8404–8414, 2020.
> >
> > **- 11/24 updated: added a missing sentence *''On the one hand, ... , as the reviewer pointed out''* in the second paragraph**

---

### Decision · Program_Chairs · 2021-01-07
**Final Decision**

**Decision:**

Accept (Spotlight)

**Comment:**

This paper advances the idea that recent “influence estimation” methods for supervised learning cannot be trivially applied to GANs. Based on Hara et al.’s method, the authors propose a novel influence estimation for GANs, and an evaluation scheme based on popular GAN evaluation methods, exploiting the fact that they are differentiable with respect to their input data. The paper demonstrates empirically that the proposed influence estimation method correlates to true influence. It also shows that removing “harmful” instances using the average log-likelihood, Inception Score, and Frechet Inception Distance versions of the proposed metric improves the quality of generated examples.

All reviewers were positive about the paper. R2 pointed out that it was well-written and appreciated the detailed analysis. They thought it thoroughly explained the similarities between it and the most closely-related recent work (Hara et al. and Koh & Liang). Concerns expressed by the reviewer were: the amount of samples needed to be removed to obtain a statistically significant result, lack of qualitative results, and an outdated baseline for anomaly detection. The reviewer also stated that they had some concerns with practical applicability and would like to see more GAN metrics, like Precision & Recall. The authors added qualitative results to the paper which partially satisfied the reviewer.

R1 also thought that the paper was well-written and contributed to the interpretability of GAN training. Like R2, they pointed out the lack of visual examples (addressed in rebuttal), and asked for more insight into what kind of characteristics make a data point influential. They also requested that the authors add a metric that trades fidelity and diversity like P&R. The reviewer originally felt that the paper was below the bar, because it was “like a story without a satisfying conclusion”. However, the authors responded with additional analysis which satisfied the reviewer, and they upgraded their score by two points.

R3 also found the paper well-written and interesting, like the other reviewers. The reviewer raised some similar concerns as the other reviewers (e.g. qualitative results), as well as the scalability of the method to relevant architectures, which I thought was surprising that the other reviewers didn’t mention. The authors responded that they believe their method succeeded in improving diversity of the generated samples but not their visual quality. This is an important point.
The additions in Appendix D have addressed the main concerns of R1 and R2, as well as R3’s concern about lack of visual analysis. R1 seems quite convinced now, and R2, though not changing their score, was already in favour of acceptance. It is an interesting finding that “harmful” instances seem to come from regions of distributional mismatch.

I would like to see a fidelity-diversity tradeoff like P&R added to a paper, and a discussion of this work in relation to DeVries et. al “Instance Selection” that appears to be similarly motivated though executed differently. I think one major thing holding back this paper is the scale of the experimental analysis (Gaussians & MNIST); I hope the authors can scale the method in future work.